# Can we infer excitation-inhibition balance from the spectrum of population activity?
Kingshuk Chakravarty[1], Sangheeta Roy[1], Aniruddha Sinha [1] & Arvind Kumar [2,3] ✉

Networks in the brain operate in an excitation-inhibition (EI) balanced state. Altered EI balance underlies aberrant dynamics and impaired information processing. Given its importance, it is crucial to establish non-invasive measures of the EI balance. Previous studies have suggested that relative EI balance can be inferred from the spectrum of the population signals such as Local Field Potentials (LFP), Electroencephalogram (EEG) and Magnetoencephalography (MEG). This idea exploits the fact that in most cases excitatory and inhibitory synapses have quite different time constants. However, it is not clear to what extent spectral slope of population activity is related to the network parameters that define the EI balance e.g. excitatory and inhibitory conductance. To address this question we simulated two different types of recurrent networks and measured spectral slope for a wide range of parameters. Our results show that the slope of the spectrum cannot predict the ratio of excitatory and inhibitory synaptic conductance. Only in a small set of simulations a change in the spectral slope was consistent with the corresponding change in the synaptic weights or inputs to the network. Thus, our results show that we should be careful in interpreting the change in the slope of the population activity spectrum.

Excitation-inhibition (EI) balance, i.e., the ratio of excitatory and synaptic conductance, has emerged as an important parameter that determines not only the stability of the network[1–3] but also the statistics of neuronal activity[4–6], and the gain of the network[7]. Experimental data have corroborated results of computational models that both timing and magnitude of excitation and inhibition are tightly maintained for brain function[8–10]. Furthermore, in vitro analysis of neuronal connectivity has revealed that EI balance may get severely altered in pathological states [see review by Tatti et al.[11]].

These observations have inspired the development of methods to estimate EI-balance non-invasively. In this regard, Gao et al.[12] have made significant progress and proposed that the aperiodic component of population activity may be used to extract information on changes in the EI balance. In particular, they argued that a steeper slope of the spectrum of the population activity (LFP/EEG/MEG) is associated with stronger inhibition (Fig. 1). This argument is based on the fact that inhibitory synapses are slower than excitatory ones and population activity is largely associated with transmembrane currents induced by synaptic inputs[13]. That is, we can consider the LFP (and EEG/MEG) spectrum to be the sum of the spectra of excitatory and inhibitory currents, and therefore, the balance of excitation and inhibition can control the slope of the spectrum. The spectral estimate of (relative) EI-balance also implicitly assumes that the inputs to the network

are both uncorrelated and asynchronous (see Discussion for further elaboration on this). However, it is well known that given the network connectivity and neuron properties, network activity may be both synchronous and oscillatory and EI-balance itself is a key determinant of the network activity state[2,3]. Therefore, it is unclear under what conditions we can use the spectral slope to compare the EI-balance between two different situations. Essentially, the question arises when and how the spectral slope is related to the network parameters.

To address this question, we simulated network activity under different levels of EI-ratio (as determined by synaptic weights and inputs) and measured the slope of the population activity spectrum. We considered two different classes of networks: a neocortical network model in which both excitatory and inhibitory populations were mutually connected, and the STN-GPe network in which excitatory recurrent connections were absent. Both network models have been extensively studied for their dynamical properties, in particular synchrony and oscillations[1–3,14–18].

We found that only in a few network configurations an increase in the inhibitory synaptic strength was associated with a corresponding increase in the spectral slope. In most cases, an increase in the spectral slope could be due to both an increase or a decrease in the strength of inhibitory inputs. In fact, an increase in the inhibition in two networks with similar parameters often resulted in a different effect on the spectral slope. It can be argued EI

¹Tata Consultancy Services, Kolkata, India. ²Division of Computational Science and Technology, School of Electrical Engineering and Computer Science, KTH Royal Institute of Technology Stockholm, Stockholm, Sweden. ³Science for Life Laboratory, Stockholm, Sweden. ✉e-mail: arvkumar@kth.se

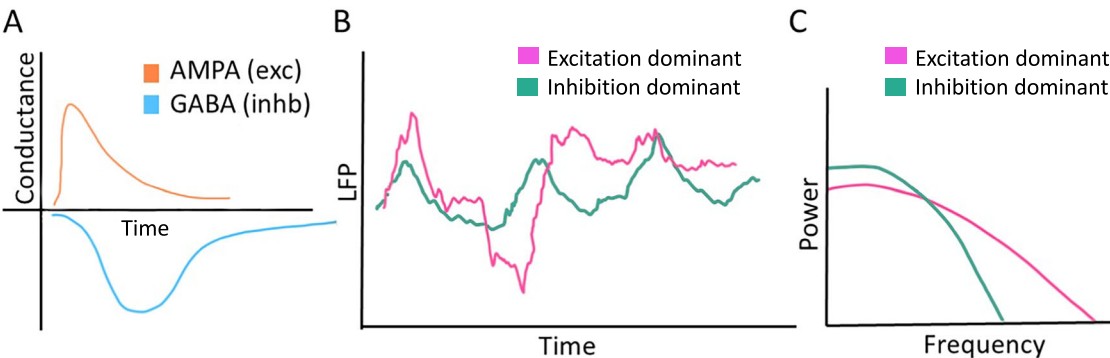

**Fig. 1 | Schematic diagram to illustrate how excitation and inhibition may influence the power spectrum. A** Schematic of excitatory and inhibitory postsynaptic conductance. **B** Schematic of LFP time series when excitation or inhibition dominates the inputs to a neuron. **C** Power spectrum of the LFP shown in panel B.

balance is a network property governed by the activity state and cannot be solely determined by individual neuron and synaptic parameters. Therefore, we checked how a change in the spectral slope was related to the ratio of total excitatory and inhibitory conductance. Even at that level, we did not find any clear relationship between the ratio of excitatory and inhibitory conductance and the spectral slope.

Thus, our results suggest that a change in the spectral slope is not a reliable predictor of EI balance in terms of network parameters or synaptic conductance. Therefore, we argue that we do not yet understand the origin of a change in the spectral slope, and we should be cautious while interpreting results based on the spectral slope.

## Results

At the level of excitatory and inhibitory neurons, there are three network motifs in the brain: (1) fully recurrent: where there are mutual connections between and within the two neuron populations, e.g., in the neocortex, (2) without E-E recurrence, e.g., in CA1 or STN-GPe network of the basal ganglia, and (3) only I-I recurrence: i.e., when excitatory neurons are not present, e.g., in the striatum. To understand when spectral slope can be used to estimate the relative balance of excitation and inhibition (in terms of network synaptic conductances or input parameters), we simulated two of the three network motifs: the STN-GPe subnetwork of the basal ganglia and a random network of mutually connected E and I neurons mimicking the neocortex. We systematically changed the EI-balance by varying the synaptic strengths and external input. We then tested when a change in the spectral slope was consistent with the EI-balance set using network parameters (inputs and synaptic strengths).

### EI-balance and spectral slope in STN-GPe network

For the STN-GPe network, we systematically varied four parameters to obtain different activity states in the network: the weight from STN to GPe, the weight from GPe to STN, and the background inputs to both STN and GPe. This resulted in 625 simulations with different parameter combinations (network activity for two such combinations shown in Fig. 2A–F, G–L). For each parameter configuration we simulated 25 trials. Thus, we obtained $OI_{avg}$ and $\gamma_{avg}$ for each of the 625 configurations over 25 trials to ensure stability of the estimate. When we rendered the data in the space spanned by $OI_{avg}$ and $\gamma_{avg}$ we observed two distinct clusters (Fig. 3 black dots). These two clusters were mainly defined by the oscillatory dynamics in the network. Therefore, based on distribution of $OI$ values (Fig. 3 black dots), we distinguished between a "low-oscillation regime" ($OI_{avg} \leq 0.4$) and a "high-oscillation regime" ($OI_{avg} > 0.6$). The thresholds for low-oscillation and high-oscillation regimes had been chosen empirically. The high oscillatory regime was characterized by a wide range of $\gamma_{avg}$, whereas in the low oscillation regime $\gamma_{avg}$ was distributed over a narrow range. Based on the distribution of $\gamma_{avg}$ we defined a "high exponent state" ($\gamma_{avg} \geq 3$) and a "low exponent state" ($\gamma_{avg} < 3$).

For each of the activity regimes we varied the inhibitory inputs to the GPe neurons and thereby changed the EI balance in the network. Thus, at the synaptic level we decreased inhibition in the STN by reducing the firing rate of GPe neurons. Such a change is known to affect oscillations[15]. Here, we found that GPe inhibition also leads to a wide range of changes in the $\gamma_{avg}$ depending on the network activity regime and network parameters (Fig. 3). To quantify these changes in $\gamma_{avg}$ and $OI_{avg}$, we measured the angle of the line joining the network state with and without GPe inhibition (e.g., red lines in Fig. 3A–C: arrowhead marks the location in the presence of GPe inhibition). An increase in $\gamma_{avg}$ and $OI_{avg}$ would result in an angle between $0°-90°$, whereas a decrease in $\gamma_{avg}$ and an increase in $OI_{avg}$ would result in an angle between $270°-0°$ (i.e., $270°-360°$).

In low $OI_{avg}$ regime for some network configurations the change in $OI_{avg}$ and $\gamma_{avg}$ was very small. However, in general inhibiting GPe resulted in an increase in $OI_{avg}$, while $\gamma_{avg}$ could either increase or decrease as summarized in the radar plots in the inset of Fig. 3A. For another set of network parameters that rendered the network in a low $OI_{avg}$ regime, we observed a much bigger change in both $OI_{avg}$ and $\gamma_{avg}$. For these network configurations, an increase in inhibition of GPe neurons could either increase or decrease $\gamma_{avg}$ (Fig. 3C). Similar results were obtained when the network was operating in the high $OI_{avg}$ regime. In this regime, an increase in $OI_{avg}$ was not much, but $\gamma_{avg}$ showed both an increase or a decrease. In all states, there was a greater tendency for a reduction in $\gamma_{avg}$ (Fig. 3D).

Thus, inhibition of GPe did not consistently reduce $\gamma_{avg}$ in the STN; however, for a substantial number of cases, it reduced the exponent (Fig. 3D)[19], as we may expect due to weaker inhibition of STN neurons by GPe. Instead, the change in the spectral exponent seems to depend on both network operating regime and network parameters.

Next, we asked whether similar network states (as defined by $OI_{avg}$ and $\gamma_{avg}$) could undergo similar transitions when GPe was inhibited, and which state variable predominantly influences the transition pattern. To this end, we varied one of the network parameters and examined its effects on oscillation and exponent values (Fig. 4). As expected, the effect of each parameter taken individually was different in the two activity regimes.

The low-oscillatory regime required high background input to the STN neurons or a weak connection from STN to GPe, while other parameters could take values from a large distribution (Fig. 4A) or set in a 'sloppy' manner. High-oscillatory state, on the other hand, required weak input to STN neurons and strong connections from STN to GPe. This is consistent with previous models[15,17]. Upon an increase in inhibitory inputs to the GPe neurons, the network could make three possible transitions: either stay in their initial low or high oscillatory state, or it could make a transition from low to high oscillation state. In each of the three transitions, the network could take a wide range of parameters, and based on a single parameter, it was not possible to predict what kind of transition would be observed (Fig. 4B).

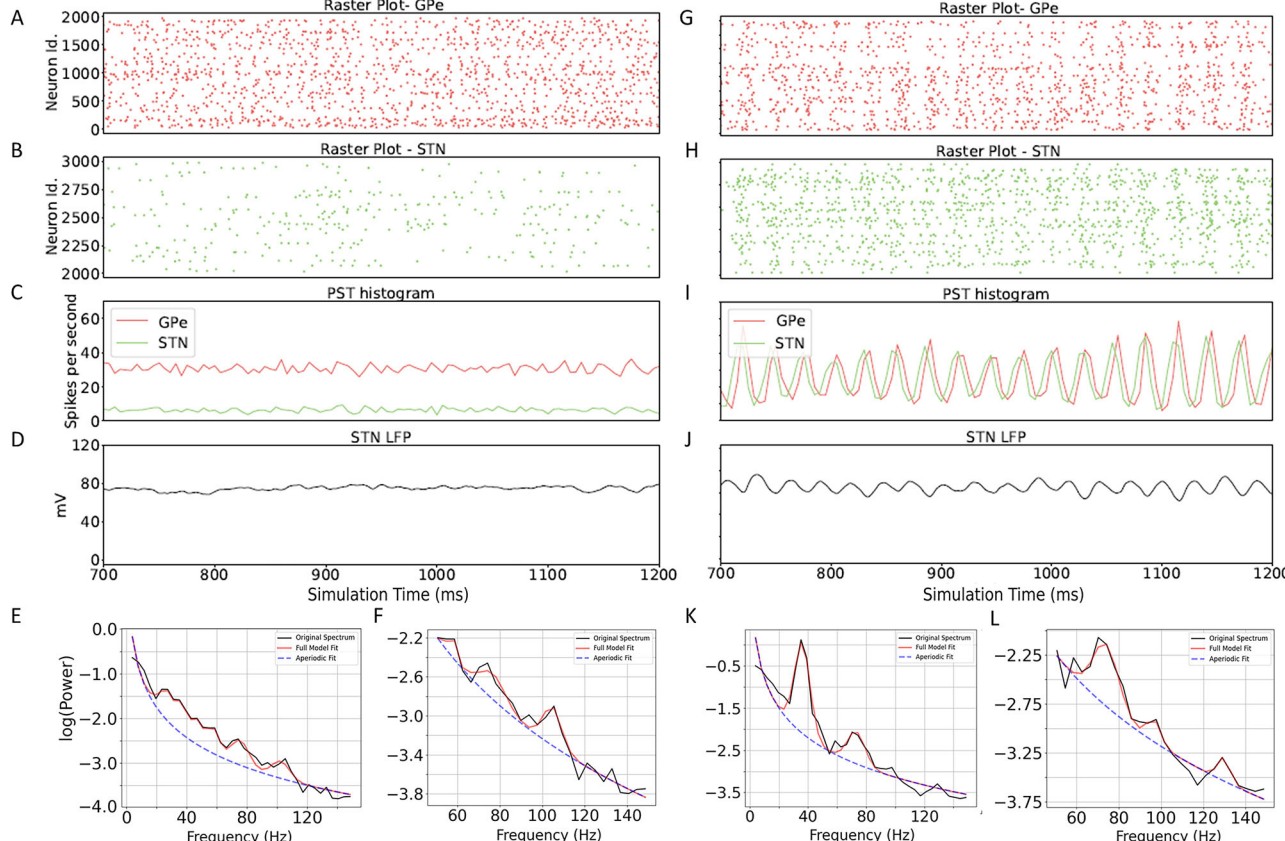

**Fig. 2 | Example of network activity and estimation of the slope of the population activity spectrum.** Computation of $\gamma$ from the total conductance for network configurations corresponding to non-oscillatory ($OI = 0.32$ and $\gamma = 3.49$: **A–F**) and oscillatory conditions ($OI = 0.576$ and $\gamma = 3.63$: **G–L**). **A, B** Raster plots of GPe and STN activities, respectively. **C** Mean firing rate of STN and GPe population. **D** LFP trace corresponding to the population activity in STN. **E, F** Power spectral density of the LFP shown in (**D**). The red line shows the fitting of PSD using FOOOF. Blue

dotted line shows the corresponding $1/f^{\gamma}$ estimate, i.e., the aperiodic component. **E** The estimation of the aperiodic component from 5–150 HZ for $\gamma$ computation. **F** estimate of the aperiodic component from 50–150 Hz. **G–L** Same as **A–F** for network configurations capable of generating oscillations in the $\beta$-range ($OI = 0.576$ and $\gamma = 3.63$). In **K**, we show an example with $\beta$-peak which is a characteristics of PD. The sample illustration is generated from data obtained from a single trial.

Finally, we looked at the change in $\gamma_{avg}$ when the network made a transition from low to high oscillation regime. In these simulations, $\gamma_{avg}$ either increased or decreased from the starting value. Once again, individual parameters were drawn from a larger distribution for these transitions. Thus, individual parameters cannot help in predicting whether the network will show an increase or a decrease in $\gamma_{avg}$ when oscillations increase (Fig. 4C). However, stronger synaptic weight from the STN to GPe was associated with increased exponent values during the transition from low to high oscillations[20].

To better understand the relationship between EI balance and the exponent of the spectrum of population activity, we analyzed the changes in $\gamma$ and $OI$ as a function of average total excitatory and inhibitory conductances received by neurons (Fig. 5). As we increased the inhibition of GPe, oscillations and $\gamma_{avg}$ could increase or decrease. To quantify this change, we defined four possible transitions, i.e., both $\gamma$ and $OI$ can increase or decrease independently. We expected that an increase in inhibitory conductance would be associated with an increase in $\gamma_{avg}$ and oscillations would increase (i.e., arrows pointing north, north-east). However, when we rendered the data as a function of E and I conductance, we did not find a clear relationship between E-I conductance ratio and direction of the arrows (see Fig. 5). For the same average excitatory-inhibitory conductance, we could both see an increase and a decrease in $\gamma_{avg}$, irrespective of whether the starting state of the network was oscillatory or non-oscillatory (compare Fig. 5A, B). The results were the same when restricting the analysis to cases when increased inhibition of GPe did not result in a change in the activity regime.

## Neocortical network model
A lack of a clear relationship between the slope of population activity spectrum and excitation-inhibition balance could be a property specific to the STN-GPe network. Therefore, next we turned to a model of a neocortical network in which both excitatory and inhibitory connections are mutually connected with a fixed connection probability (see "Methods").

In this model, we systematically varied the external input, the ratio of recurrent inhibition and excitation, and the ratio of NMDA and AMPA components in excitatory synapses. These parameters rendered the network with different activity states characterized by weak and strong oscillations (also synchrony). For each combination of network parameters, we estimated the strength of oscillations ($OI_{avg}$) and the exponent of the spectral slope ($\gamma_{avg}$). As is already known, oscillations increased with the strength of external inputs (Fig. 6A). By contrast, the effect of increasing inhibition was contingent on the external input. When the network was driven by strong external input, increasing inhibition tended to reduce oscillations (Fig. 6A). Increasing the NMDA component in excitatory synapses increased oscillations without affecting the dependence of oscillations on $g$ and $\eta$.

Unlike the oscillations, the exponent of spectral slope ($\gamma_{avg}$) did not show any pattern with respect to $g$ and $\eta$ (Fig. 6B). An increase in the NMDA component made the synapses slower. Therefore, increase in the NMDA component resulted in an increase in the spectral slope (compare rows of the Fig. 6B).

To further illustrate that $\gamma_{avg}$ is not directly related to EI balance in terms of synaptic conductances, we compared how $\gamma_{avg}$ and $OI_{avg}$ change when synaptic inhibition was increased from a baseline state. For the

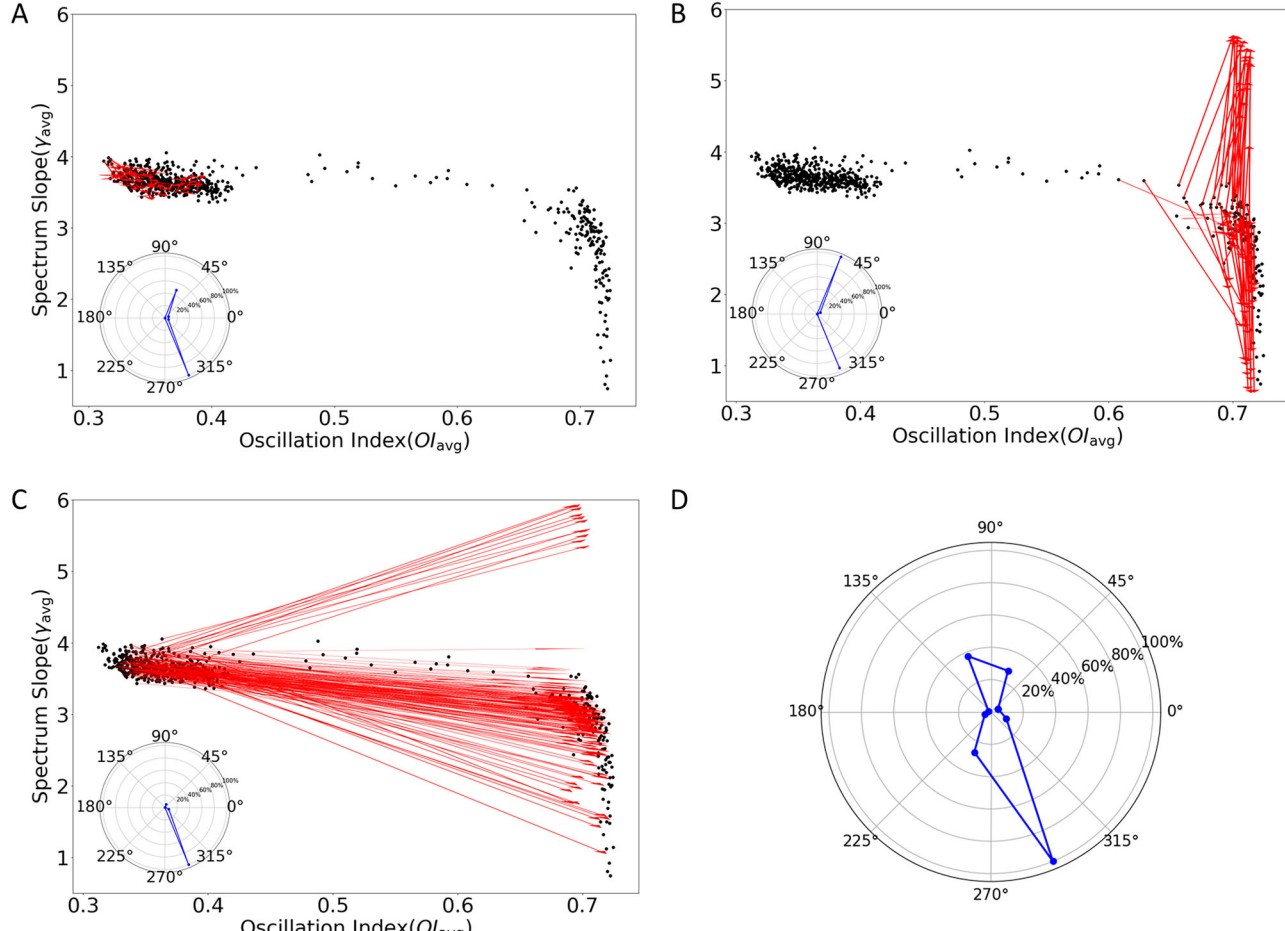

**Fig. 3 | Change in STN-GPe network states when GPe was inhibited. A** Black dots indicate the state of the STN-GPe network as defined by $\gamma_{avg}$ and $OI_{avg}$. Each black dot corresponds to one of 625 configurations of the network parameters. Red arrow heads indicate the change in the network state when GPe was inhibited. Here red arrows show only the transitions for network configurations that resulted in a non-oscillatory regime and GPe inhibition did not induce an activity regime change. Inset: distribution of directions of red arrows. **B** Same as in **A** but red arrows here indicate inhibition induced state change for networks that were operating in a high oscillation regime. **C** Same as in **A** but red arrows here indicate inhibition induced state change for networks that were initially operating in a low oscillation regime. **D** Histogram of the transition directions of $[OI_{avg}, \gamma_{avg}]$ occurring with 600 Hz inhibitory input in the network for all configurations.

baseline state, we chose $g = 4$. We then increased $g$ and rendered the network state in the space spanned by $\gamma_{avg}$ and $OI_{avg}$ (Fig. 7A, B). Because we increased $g$ we expected $\gamma_{avg}$ to increase. That is, we expect the arrows in Fig. 7A, B to point north or northeast direction. However, as is evident in Fig. 7A, B increases in $g$ could both increase or decrease $\gamma_{avg}$ (see summary in Fig. 7C).

Thus, our simulations show that in both the STN-GPe network and a simplified model of the cortical network, the slope of the population activity spectrum is not changing in a manner consistent with the network parameters (synaptic weights and inputs).

## Discussion

Balance of excitation-inhibition is an important descriptor of the dynamics[1–3], and function of a network[4,21]. Gao et al.[12] have suggested that relative EI-balance information can be inferred by comparing the slope of the spectrum of population activity, such as EEG/MEG. This has renewed an interest in measuring the slope of the LFP/EEG/MEG spectrum in various behavioral and disease states[22]. Essentially, Gao et al.[12] implicitly suggested that there is a systematic relationship between the spectral slope and the determinants of EI balance in a network (i.e., network parameters). Here, we set out to test this idea in two different network models where we knew the ground truth in terms of the conductance of excitatory and inhibitory synapses. We found that only in some cases a change in the spectral slope

was consistent with the change in the EI-balance as determined by our network parameters.

In general, in our simulations, it was not possible to say that a state with a higher $\gamma_{avg}$ was generated by a network with stronger inhibitory conductances (due to inputs or synaptic weights) than the other state with a smaller $\gamma_{avg}$. In our model, we varied the network parameters in a range wide enough to produce different activity regimes (e.g., oscillatory or non-oscillatory). However, even $g$ has no clear relationship with the spectral slope (Fig. 6). Thus, in both network models, we did not find any systematic relationship between spectral slope and network parameters. This is a rather surprising observation and suggests that changes in the spectral slope cannot always be interpreted in terms of putative changes in the network parameters.

When we consider an unconnected population of neurons driven by excitatory and inhibitory spikes, indeed, the spectral slope is determined by the ratio of excitatory and inhibitory inputs. This follows from Campbell's theorem[23], given the fact that a neuron can be considered to be a linear time-invariant system. However, in a recurrent network, as we change the excitatory drive or recurrent inhibition, a recurrent network responds with a change in the firing rates of both excitatory and inhibitory neurons—the exact change in the firing rates depends on the network configuration (balanced or inhibition stabilized mode) and strength of synapses[1,24,25]. Moreover, network interactions make the activity less regular or bursty than

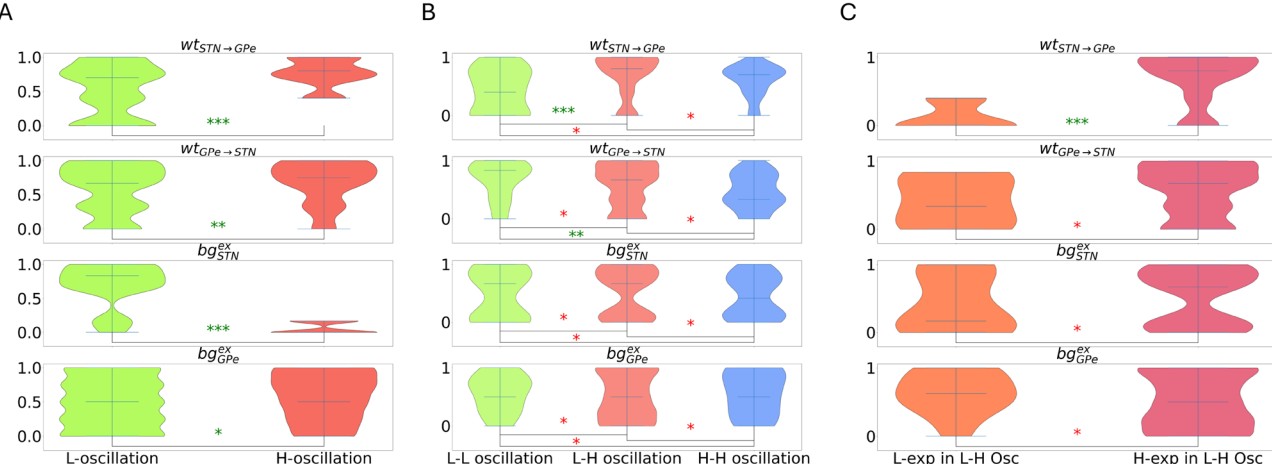

**Fig. 4 | Effect of network parameters in shaping the network state and its transition when GPe is inhibited.** For a meaningful comparison we normalized each parameter using the formula $\frac{x-\min(x)}{\max(x)-\min(x)}$, where $x$ comprises $(STN-GPe)_{wt}$ or $wt_{STN-GPe}$, $(GPe-STN)_{wt}$ or $wt_{GPe-STN}$, $STN_{bg}$ or $bg_{STN}^{ex}$, $GPe_{bg}$ or $bg_{GPe}^{ex}$. **A** Distribution of four parameters when the network was either in low-oscillation or high-oscillation regimes. This refers to the states marked with black dots in the Fig. 3A–C. **B** The distribution of parameters during various types of inhibition-induced transitions,

namely low to low, low to high, and high to high oscillation zones. **C** Same as in **B** Distribution of parameters when GPe inhibition resulted in transition to a smaller and higher exponent value from their baseline. This panel was made using data shown in Fig. 3C. "*" denotes significance level (p value): green "***" for $p < 0.001$, green "**" for $p < 0.01$, green "*" for $p < 0.05$. A red "*" indicates a non-significant result ($p > 0.05$).

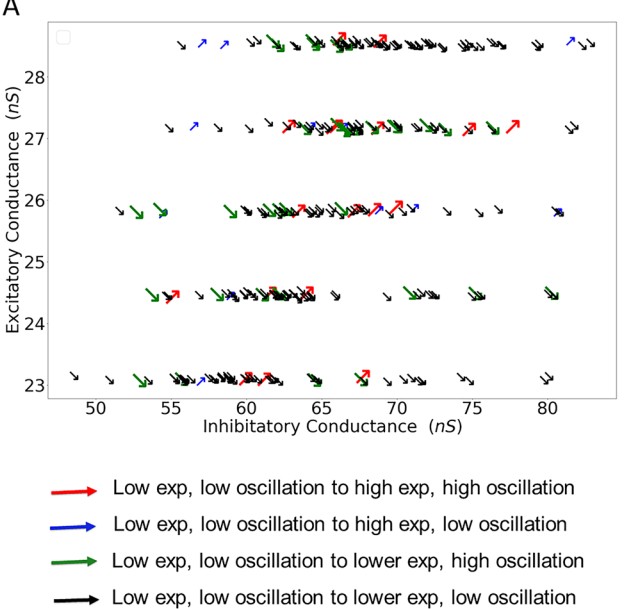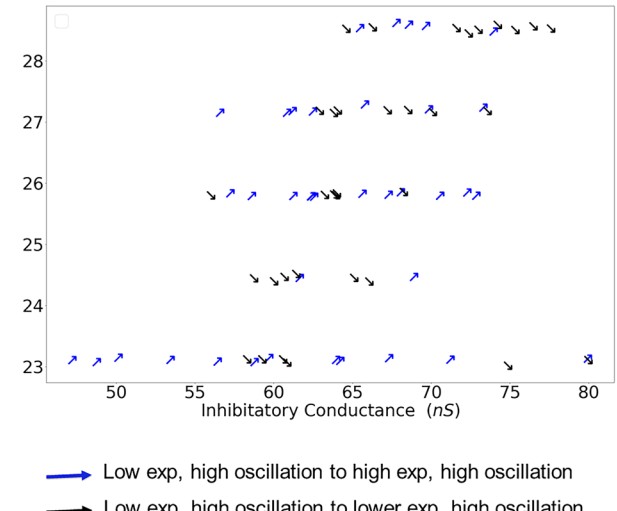

Low exp, low oscillation to high exp, high oscillation

Low exp, low oscillation to high exp, low oscillation

Low exp, low oscillation to lower exp, high oscillation

Low exp, low oscillation to lower exp, low oscillation

Low exp, high oscillation to high exp, high oscillation

Low exp, high oscillation to lower exp, high oscillation

**Fig. 5 | Relationship between excitatory and inhibitory conductance and the slope of the population activity spectrum.** Here we re-render the data shown in Fig. 3. Inhibition induced changes in the network state were reduced to four possibilities: increase or decrease in oscillations and increase or decrease in spectral slope. Thus, the network could move in four possible directions. These directions are rendered as

arrows in the two panels as a function of average excitatory (y-axis) and inhibitory conductance (x-axis). **A** Inhibitory induced change in the network state when the network was operating in a non-oscillatory regime. **B** Same as in **A** but here the network was operating in an oscillatory regime. In both cases we did not find a clear trend between direction of the arrows and average excitation-inhibition.

a Poisson process[2,3]. This implies that the spectrum of the inputs to the neurons has peaks. Finally, network interaction makes the activity correlated[2,3], which can further amplify the excitatory or inhibitory component of the synaptic current flowing through the membrane (see Supplementary Fig. S1).

Thus, the exact shape of the population activity spectrum depends on the activity regime, network structure, neuron, and synapse properties. When the network is tuned to operate in an asynchronous-irregular state, the spectral power of the noise is by definition $1/f^a$, where $a > 0$. When the network is operating in an oscillating regime, additional peaks are added in the $1/f^a$ spectrum. In some cases, when neurons have voltage-gated channels,

it is possible that neurons do not behave as low-pass filters. In such a setting, population activity generated by the network may have significant deviations $1/f^a$ spectrum. To the best of our knowledge, no one has shown the existence of noise in which spectral power increases with frequency, i.e., $a < 0$ (e.g., blue noise) in neural networks.

Unless excitatory and inhibitory inputs are tightly balanced, the effects of neuron/synapse properties, network activity regime (spike time regularity and synchrony) cannot be removed from the spectrum of membrane conductance or LFP. Therefore, it is not a surprise that a change in the spectral slope does not indicate a change in excitatory and/or inhibitory input rates or synaptic strengths.

**Fig. 6 | Network oscillations and slope of the spectrum of population activity in a model of neocortical network. A** Oscillation ($OI_{avg}$) and **B** slope of the spectrum ($\gamma_{avg}$) as a function of ratio of recurrent inhibition and excitation ($g$) and external input ($\eta$). Each row represents the network with a specific ratio of NMDA and AMPA current in excitatory synapses.

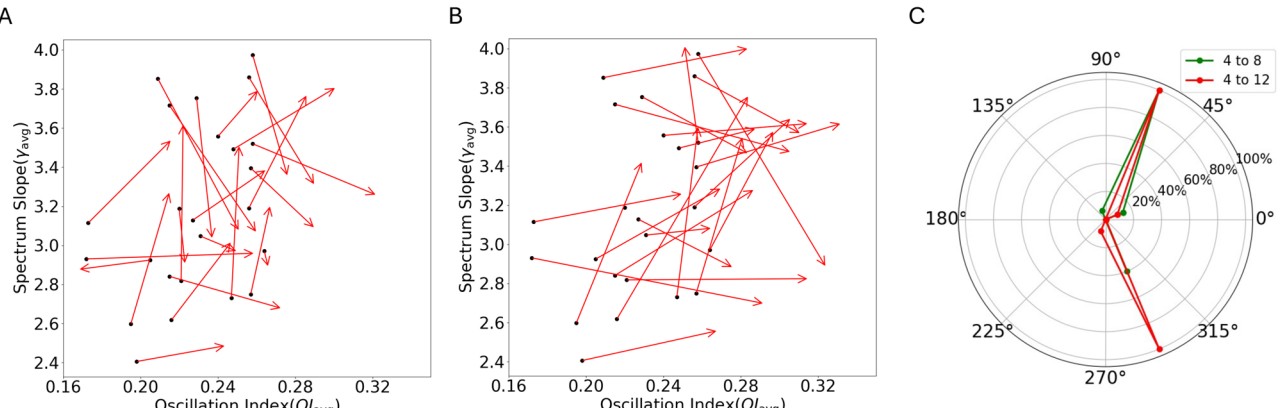

**Fig. 7 | Increase in synaptic inhibition does not imply an increase in $\gamma_{avg}$. A** The starting points of the arrows show the network state ($OI_{avg}$, $\gamma_{avg}$) for $g = 4$ and arrows heads indicate the network state for $g = 8$. **B** Same as in **A** but now arrows heads indicate the network state for $g = 12$. **C** Distribution of the directions of the arrows shown in (**A**) and (**B**). Red trace: distribution for change in network state when $g$ changes from 4 to 8. Green trace: distribution for change in network state when $g$ changes from 4 to 12.

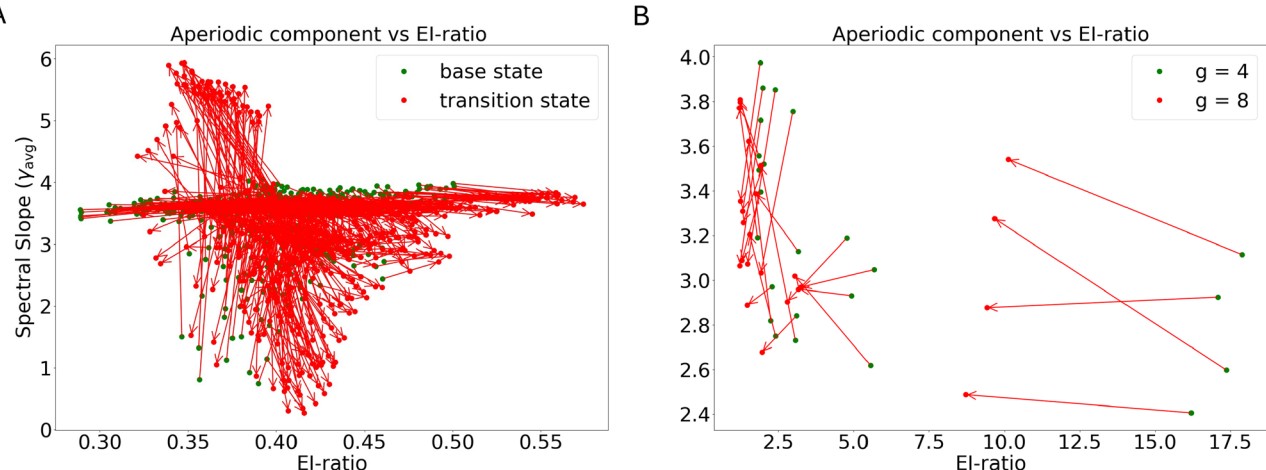

**Fig. 8 | Analysis of the relationship between the EI-ratio and the exponent values—derived using the FOOOF algorithm applied to the PSD of the total conductance time series. A** For STN-GPe network. Green dots represent baseline states (without inhibition), while red dots correspond to network outputs under 600 Hz inhibitory input. **B** Same as in **A** but for the neocortical network.

To further illustrate this, we rendered spectral slope against the ratio of mean excitatory (E) and inhibitory (I) conductances for each network (Fig. 8). Both, in the baseline state of the network (without additional inhibitory input or increase in g) there is no consistent or monotonic relationship between the EI-ratio and the exponent values (green dots in Fig. 8A, B). An explicit increase in inhibitory drive (STN-GPe network) or inhibitory conductance (g in the neocortical network) also did not produce any systematic correlation between these variables (red dots in Fig. 8A, B).

However, in certain parameter configurations, exponent values decreased with increasing EI-ratio (lines with negative slope in Fig. 8A, B), suggesting that under specific network states or parameter regimes, a relationship between the EI-ratio and the exponent may emerge.

Recently, two studies have used computational models similar to our cortical model to test how spectral slope relates to the EI-ratio in a random network[26,27]. These two studies concluded that spectral slope increases with an increase in the strength of inhibitory synapses. By contrast, in our simulations, we observed a change in the spectral slope consistent with a change in the EI-ratio (arrows pointing in the bottom left corner in Fig. 8) for a small subset of parameters. Given our results, it seems that the previous models[26,27] were tuned to operate in specific parameter and activity regimes where the relationship between EI-ratio and spectral slope was consistent. Thus, a spectral slope is an indicator or EI-ratio for specific activity regimes, e.g., when network activity is asynchronous and irregular, such as spike regularity and correlation do not 'contaminate' the spectrum of excitatory and inhibitory currents.

Admittedly, the network models we have considered are highly simplified (e.g., they consist of a homogeneous neuron population connected with the same synaptic strengths). In particular, such network models do not generate a biologically realistic spectrum of population activity—in the sense that such models often produce a single frequency oscillation. Furthermore, we have used point neurons in the model, and therefore, our model of LFP cannot capture the full nuances of the LFP (or our such measures of population activity)[28]. Similarly, as we have used point neurons, we could only generate LFP by summing synaptic currents. It is not feasible to use morphologically realistic neuron models in network simulations. However, the literature suggests that networks with morphologically realistic neurons model also produce similar dynamics[29] and neuron shape by itself does not alter the shape of population activity waveforms[30]. In addition, we have also ignored the role synaptic delays may play in shaping the EI-ratio and spectral slope. We and others have shown that synaptic delays can introduce complex dynamics and population activity spectrum[31,32], and therefore the relationship between EI-ratio and spectral slope is likely to be rather complex if we also vary synaptic delays. Our claim here is that if we cannot find a

relationship between network parameters and spectral slope even in a simplified model like ours, it is unlikely that a relationship will show up in more biological models. While both spectral slope and EI balance are determined by network parameters and network dynamics, they perhaps represent different kinds of coarse-graining, and therefore, they are not related directly in general.

Irrespective of the simplicity of our models, the message is clear that EI balance is a very complex concept and it emerges in the network through interactions of excitatory and inhibitory neurons, given synaptic strengths, inputs, and connectivity. Therefore, even if we notice a change in the synaptic strengths, we cannot predict which way the EI balance (and spectral slope) would shift or the network will restore the balance. That is, a priori, we should not assume that an increase in the spectral slope implies an increase in the inhibition (or NMDA component in excitatory synapses).

Given the importance of estimating EI-balance from the population activity spectrum, it is crucial that spectral slopes are measured experimentally under different behavioral states and stimulation paradigms to develop a more correct interpretation. We also note that even if it is not possible to relate the spectral slope to network parameters (and EI-balance), as long as the measure forms a basis for the classification of different patient groups, the measure remains valuable.

## Methods
### Network structure
**STN-GPe network.** Here, we used a previously published reduced model of the STN-GPe subnetwork of the basal ganglia[16] and modified the inputs and connectivity strengths. Briefly, the network consisted of one excitatory and one inhibitory population, which represent the STN and GPe subnuclei of the basal ganglia, respectively (Fig. 9). STN neurons received inhibitory (GABAergic) connections from the GPe, while the GPe received both excitatory (glutamatergic: AMPA) connections from the STN and inhibitory (GABAergic) connections from within the GPe itself. Neurons were connected randomly with a fixed connection probability[15] with conductance-based synapses (see Table 1; for more details, please see ref. 16). In simulations, we set the in-degree of each neuron to be fixed according to the connection probability, and the out-degree of neurons followed a binomial distribution.

STN and GPe neurons were simulated using leaky integrate-and-fire (LIF) neuron models (see Tables 2 and 3 for model parameters). Neurons were driven by Poisson-type excitatory spike trains to achieve an in vivo level of spiking activity. We varied the STN-GPe dynamics by changing following four variables: rate of background inputs to STN and GPe neurons and the strength of synaptic connections STN (i.e., STN → GPe,

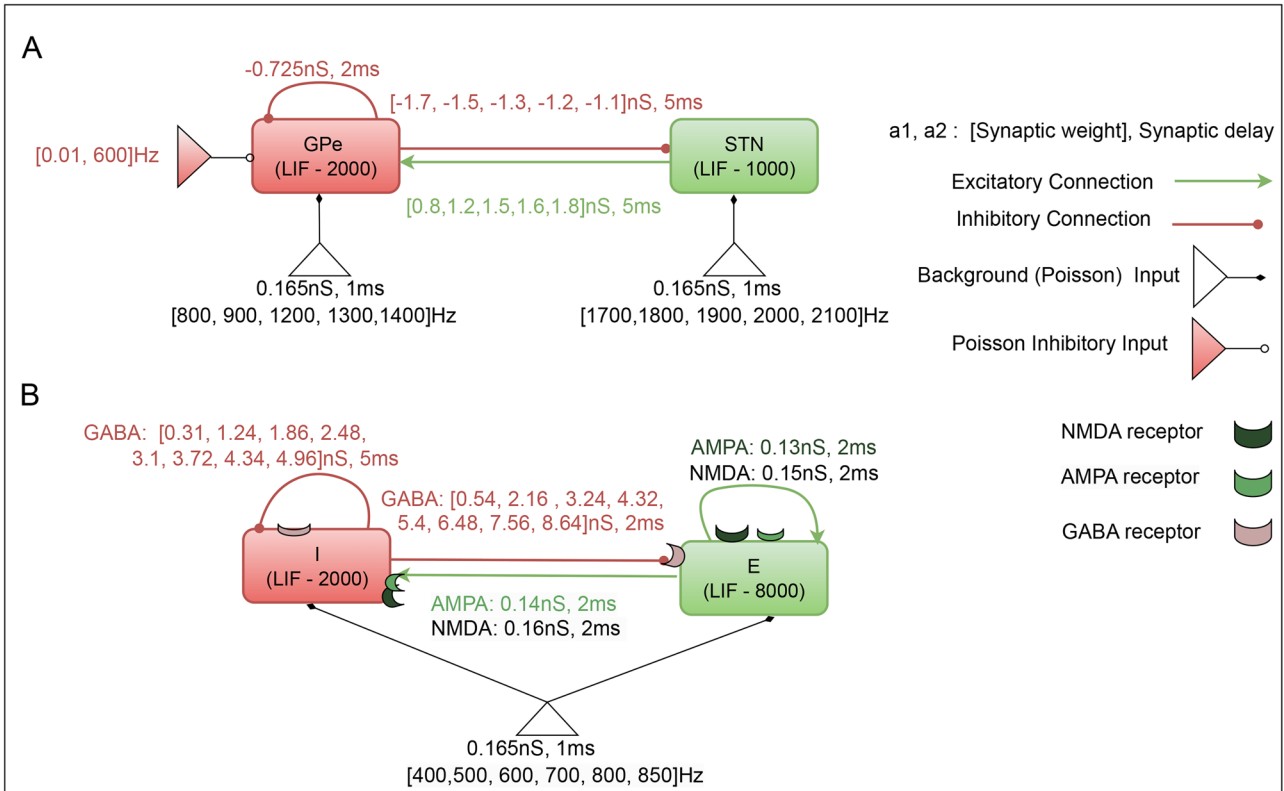

**Fig. 9 | Schematic diagram of the two networks. A** STN-GPe network. **B** A network with E → E, E → I, I → E and I → I connections. Each neuron in a population (rectangles) received Poisson type excitatory (empty triangles) and inhibitory (filled triangles) spiking input. Strength, synaptic delay and range of input rates of the inputs are written just next to the triangles. Strength and delays of the recurrent connections are written next the relevant connectors (arrows: excitatory and filled circles: inhibitory).

## Table 1 | Synaptic connection properties of the STN ↔ GPe network model

| Synaptic connection | No. incoming synapses/ neuron | Weights | Delay (in ms) |
|---|---|---|---|
| STN → GPe | 20 | 1.5 | 5 |
| GPe → STN | 40 | −1.3 | 5 |
| GPe → GPe | 40 | −0.725 | 2 |

Connection probabilities were based on values reported by Kumar et al.[15].

## Table 2 | STN and GPe neuron parameters (leaky integrate-and-fire (LIF) model)

| Name | Value | Description |
|---|---|---|
| $V_{reset}$ | −70 mV | Reset value for $V_m$ after spike |
| $V_{th}$ | −54 mV | Spike threshold |
| $tau_{syn}^{ex}$ | 5 ms | Time constant of excitatory synaptic conductance |
| $tau_{syn}^{in}$ | 10 ms | Time constant of inhibitory synaptic conductance |
| $E_{ex}$ | 0 mV | Excitatory reversal potential |
| $E_{in}$ | −80 mV | Inhibitory reversal potential |
| $C_m$ | 200 pF | Membrane capacitance |
| $g_L$ | 10 nS | Leak conductance |
| $t_{ref}$ | 5 ms | Duration of refractory period |

## Table 3 | Parameters of the neocortical network model

| Name | E | I | Description |
|---|---|---|---|
| $V_{reset}$ | −65 mV | −65 mV | Reset value for $V_m$ after spike |
| $V_{th}$ | −54 mV | −54 mV | Spike threshold |
| $E_L$ | −70 mV | −70 mV | Leak reversal potential |
| $E_{ex}$ | −10 mV | −10 mV | Excitatory reversal potential |
| $E_{in}$ | −84 mV | −84 mV | Inhibitory reversal potential |
| $C_m$ | 60 pF | 60 pF | Membrane capacitance |
| $g_L$ | 12 nS | 12 nS | Leak conductance |
| $t_{ref}$ | 2 ms | 2 ms | Duration of refractory period |
| $\tau_{syn}^{in}(GABA)$ | 5 ms | 5 ms | Time constant of inhibitory synaptic conductance for GABA |
| $\tau_{syn}^{ex}(NMDA)$ | 30 ms | 30 ms | Time constant of excitatory synaptic conductance for NMDA |
| $\tau_{syn}^{ex}(AMPA)$ | 3 ms | 3 ms | Time constant of excitatory synaptic conductance for AMPA |

GPe → STN). To mimic pathological oscillations observed in Parkinson's disease, we provided additional inhibitory input (Poisson distributed spike trains) to GPe neurons[15].

**Neocortical network.** For a model of cortical network, we simulated a network of excitatory and inhibitory neurons in which both populations were mutually connected with fixed connection probability[3] (see Table 4). Similar to the STN-GPe model, in the neocortical network model, both the excitatory pyramidal (E) and inhibitory interneurons (I) were

**Table 4 | Synaptic connection properties of the neocortical network model**

| Connection | Incoming synapses/neuron | Weights | Delay (in ms) |
|---|---|---|---|
| $(E \rightarrow I)^{NMDA}$ | x | 0.16 | 2 |
| $(E \rightarrow I)^{AMPA}$ | 800-x | 0.14 | 2 |
| $(E \rightarrow E)^{NMDA}$ | x | 0.15 | 2 |
| $(E \rightarrow E)^{AMPA}$ | 800-x | 0.13 | 2 |
| $I \rightarrow E$ | 200 | 0.547 | 2 |
| $I \rightarrow I$ | 200 | 0.31 | 5 |

Here, x was varied from 0 to 500 with an increment of 50. Connection probabilities were based on values reported by Kumar et al.[3].

modeled using LIF neurons and connected using conductance-based synapses. Deviating from commonly used models, we endowed the neurons with different neurotransmitters: excitatory neurons released both AMPA and NMDA, while inhibitory neurons released GABA.

**Synapse model.** For both the above networks, neurons were connected using conductance-based synapses, i.e., each incoming spike created an alpha function (Eq. (1)) shaped conductance transient.

$$G_{syn} = g_{max}^{syn} \frac{t}{\tau_{syn}} \exp(-\frac{t}{\tau_{syn}}), \text{ where syn} \in \{AMPA, \ NMDA, \ GABA\}$$

(1)

$g_{max}^{syn}$, (referred to as synaptic weight) is the maximum conductance elicited by an incoming spike.

Such a conductance transient led to a synaptic current depending on the difference between the postsynaptic membrane potential and the synapse's reversal potential:

$$I_{syn} = G_{syn}(V_m - E_{syn})$$

(2)

where $E_{syn}$ is the reversal potential of the synapse.

To model AMPA and NMDA-type synaptic dynamics in the neocortical model, we used the multi-compartment neuronal model (iaf_cond_alpha_mc from NEST simulator v3.6.0). NEST's multi-compartmental framework allows for compartment-specific tuning of parameters such as synaptic time constants and conductances for both excitatory and inhibitory inputs. We leveraged this feature to model the receptor-specific synaptic dynamics. Different ports were assigned to NMDA, AMPA, and GABA receptors, enabling synapses to be categorized based on their input type, i.e., inhibitory GABAergic or excitatory glutamatergic. GABAergic synapses were assigned to the soma, while glutamatergic synapses were connected to the NMDA and AMPA receptor ports. In our implementation, AMPA and NMDA ports were assigned to proximal and distal dendritic compartments, respectively. The synaptic time constants, leak conductance, and capacitance values were tuned to ensure that each pyramidal neuron maintained a firing rate of about 0.68 Hz, while also ensuring that AMPA synapses exhibited faster dynamics compared to NMDA synapses. Additionally, the number of inputs targeting NMDA and AMPA receptors was a free variable in our simulation-based analysis (see Table 4).

Note that we have not implemented the Mg block in the NMDA synapse dynamics. We are using point neurons and given the inputs the membrane is always depolarized to be close to the spike threshold. In such a setting, some $Mg^{2+}$ block will be removed and NMDA currents will be there for each incoming spike. Therefore, given the point neuron model, the simplification makes sense. However, when synapses are on dendrites far away from the soma then they have a different membrane potential than soma and the $Mg^{2+}$ block has to be explicitly modeled. In the model used here adding $Mg^{2+}$ block will not affect the results.

**Table 5 | Network parameters used in STN ↔ GPe network model**

| Parameter | Value | Description |
|---|---|---|
| $STN_{bg}$ | [1700, 1800, 1900, 2000, 2100] | the external background (Poisson) input to STN |
| $GPe_{bg}$ | [800, 900, 1200, 1300, 1400] | the external background (Poisson) input to GPe |
| $(STN-GPe)_{wt}$ | [0.8, 1.2, 1.5, 1.6, 1.8] | the weights of STN-GPe |
| $(GPe-STN)_{wt}$ | [−1.7, −1.5, −1.3, −1.2, −1.1] | the weights of GPe-STN |

**Neuron model.** We used the leaky integrate-and-fire (LIF) model to simulate neurons in the network. The membrane potential ($V_m$) of a LIF neuron changed over time according to the following equation:

$$C_m \frac{dV_m}{dt} = -g_L(V_m - V_{rest}) + I_{syn}^{tot} + I_{ext}$$

(3)

Where $g_L$, $C_m$, $I_{ext}$ are membrane leak conductance, capacitance, and external input current to the neuron, respectively. $V_{rest}$ is the resting membrane potential, respectively. $\frac{dV_m}{dt}$ is the rate of change of membrane potential with respect to time. The term $I_{syn}^{tot}$ represents the sum of all synaptic inputs. In the case of neocortical it refers to the sum of AMPA, NMDA and GABA currents and in the case of STN-GPe network it refers to the sum of AMPA and GABA currents.

**Simulation details**

STN ↔ GPe network: As described in the preceding sections, the STN-GPe loop was governed by four state variables: the external background (Poisson type) input to STN ($STN_{bg}$) and GPe neurons ($GPe_{bg}$), as well as the weights of STN-GPe ($(STN-GPe)_{wt}$) and GPe-STN ($(GPe-STN)_{wt}$). We systematically adjusted these variables one by one while keeping the others constant to produce various network states. This procedure was carried out for both conditions (i.e., with and without inhibitions). The range of 4D parameters (network state variables) is outlined in Table 5. In addition, for each of these 4D network parameter configurations, we varied the inhibitory input to the GPe neurons (to mimic striatal input) to externally alter EI balance for each network state.

Various values of the 4D parameters gave rise to a variety of network activity states, distinguished by the specific oscillatory dynamics within those activities. The simulation was conducted for each network configuration spanning 1700 ms and repeated for 25 trials. The first 700 ms were used to allow the network to stabilize, and only the last 1000 ms were used for further processing and analysis.

Neocortical network model: Our experimental design for the neocortical network involved a systematic adjustment of (1) the rate of external excitatory background (Poisson) input ($\eta$) to both pyramidal and interneuron populations, (2) the fraction of excitatory synapses exclusively targeting NMDA receptors ($\zeta$, where $\zeta \in (50 \times i)$ where $i \in [0,10]$ out of 800 synapses) within the pyramidal populations, (3) the ratio ("g") of inhibitory to excitatory synaptic weights. "g" controlled the level of inhibition in the network. This 3D parameter set (configurations) [$g$, $\eta$, $\zeta$] enabled us to explore the effects of these parameters on the network activity and to elucidate their role in shaping oscillation dynamics and the excitatory-inhibitory balance of the network. Here, we simulated 25 trials for every parameter configuration and took the average of the results. Network activity analyses were carried out in two distinct network conditions.

Base state: this condition represents the baseline or control state of the network, where the parameter "g" and $\zeta$ were set to 4 and 400, respectively.

Transition state: in this condition, we systematically varied the parameter "g" to induce transitions in the network dynamics.

**A note on model parameter values.** For both models, we have taken neuron and synapse parameters from published models. It has to be

noted that in most cases, data about synaptic strength is not available as often, in experimental settings, synaptic strength is measured in vitro. However, in vivo synaptic strengths could be quite different from those in vitro due to the high conductance state of neurons. Moreover, the simulation is a scaled model of the brain networks. Therefore, following the standard practice in computational modeling of the brain networks, we tuned the parameters to get the network operating in an activity range that is consistent with the experimental measurement, e.g., firing rates, population oscillations, and synchrony. Once we got to a physiological range of network activity, we varied the synaptic strengths and inputs to test the parameter sensitivity of the results.

**Simulation tools**. The dynamics of the STN-GPe network and the cortical network were simulated using NEST (version 3.6.0)[33] with a simulation resolution of 0.1 ms. The Runge-Kutta method with a time step of 0.1 ms was used to solve the differential equations. The spiking activity, power spectrum, and exponent of the network were analyzed using custom code developed with the SciPy and NumPy libraries, which provide powerful tools for numerical computations and data analysis in Python 3.8.10.

**Data analysis**
LFP model: for network activity analysis, we recorded the excitatory and inhibitory conductances from 10 excitatory neurons (STN neurons from the STN-GPe network and pyramidal neurons from the neocortical network). These summed conductances were converted to current and treated as LFP signals[34]. Membrane conductance was recorded with a sampling rate of 1 KHz.

Spectrum of population activity: the analysis of population activity spectrum (LFP) was performed using the "Welch" method. This technique involved segmenting the signal into overlapping sections, calculating the power spectrum density (PSD) estimate for each segment, and subsequently averaging these estimates to achieve a more refined spectrum. For this, we segmented the LFP signal into overlapping (50% overlap) segments of length $M$, where $M$ (=256), was chosen based on the signal characteristics and the desired frequency resolution. Each segment was convolved with an "Hanning" window function to minimize spectral leakage and reduce artifacts. Subsequently, we applied the Discrete Fourier Transform (DFT), computed the squared magnitude of the DFT for each segment and averaged these across all segments to estimate the PSD. Finally, the spectrum of population activity was computed by averaging the PSD estimates across all frequency bins.

Oscillation index: to measure oscillation strength without explicitly knowing the dominant oscillation frequency, we first measured the entropy of the PSD. To this end, we first normalized PSD to have a unit area so that we can treat it as a probability distribution. Then we measured Shannon Entropy of the PSD-derived distribution.

$$H_s = -\frac{\sum_{i=1}^{N} P(f_i)\log_2 P(f_i)}{\log_2 N} \quad (4)$$

where $P(f_i)$ is the normalized power at frequency bins ($f_i$) and $N$ is the total number of frequency bins considered. $H_s$ is smaller when power is concentrated in a few frequencies or narrow band and maximum when power is uniformly distributed over the whole spectrum. That is, the stronger the oscillations, the smaller the $H_s$. Therefore, we defined the oscillation index (OI) as:

$$OI = 1 - H_s \quad (5)$$

Estimation of the exponent of the LFP spectrum: the decay of the spectral power of LFP-like signals as a function of frequency follows a power law (i.e., $1/f^\gamma$). Here we used the FOOOF algorithm[35] to estimate the exponent $\gamma$ (see Eq. (6))

$$P(f) = \frac{a}{f^\gamma} + b, \quad (6)$$

where $P(f)$ is the power spectral density at frequency $f$. $\gamma$ reflects the frequency-dependent decay of aperiodic activity, with the exponent $\gamma$ determining its slope. We configured the FOOOF algorithm parameters as follows: maximum number of allowed peaks in the PSD within the 50–150 Hz range (post-$\beta$ range): 16; minimum peak height set to 0.01; peak threshold set to 2; peak width limits set to [0.5, 12.0] and the aperiodic mode was fixed. The lower bound of 50 Hz was chosen to minimize the impact of low-frequency oscillations in the network, which could result in error-prone aperiodic fits, while the upper bound 150 Hz was set to avoid spectral plateaus[36]. FOOOF was able to fit the data well, with an $R^2$ of $0.94 \pm 0.026$ for the STN-GPe network and $0.971 \pm 0.0035$ for the neo-cortical network, showing that it can reliably estimate the aperiodic components ($\gamma$).

**EI-ratio**
We define EI-balance as the ratio of average excitatory to average inhibitory conductance. To this end, we recorded total excitatory and inhibitory conductance of 10 neurons for 1700 ms. Ten neurons may be too small to generate LFPs. But our model is homogeneous, therefore adding more neurons does not affect the LFP properties. For a few cases, we compared LFP generated by 10 neurons or all the neurons in the network—LFPs in both cases had the same properties (see Supplementary Fig. S2).

**Statistics and reproducibility**
We do not rely on statistical testing for the key results. To reproduce the results, we have provided the simulation code on GitHub (see above).

**Reporting summary**
Further information on research design is available in the Nature Portfolio Reporting Summary linked to this article.

**Code availability**
The code to simulate the network is available on GitHub (https://github.com/arvkumar/ei_balance_lfp_spectrum).

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

## Acknowledgements
We thank Dr. Pascal Helson and Archishman Biswas for helpful suggestions and feedback on the text.

## Author contributions
A.K.: designed and conceptualized the study, supervised the work, and wrote the paper. K.C.: wrote the code to simulate the model, analyzed the data, and wrote the paper. S.R.: wrote the code to simulate the model, analyzed the data, and wrote the paper. A.S.: designed the study, analyzed the data, and wrote the manuscript.

## Funding

## Competing interests
The authors declare no competing interests.
