## [Transparent Peer Review file · Communications Biology]

Can we infer excitation-inhibition balance from the spectrum of population activity?

Corresponding Author: Professor Arvind Kumar

Version 0:

Reviewer comments:

Reviewer #1

(Remarks to the Author)

In this study, Chakravarty and colleagues investigate whether the spectral slope of aperiodic activity of a power spectrum density (PSD) can indicate the excitation-inhibition ratio in neural networks, with a steeper slope indicating higher inhibition. To explore this, they use two spiking neural network models—one representing the basal ganglia and the other an arbitrary cortical area. They systematically vary the level of inhibition in the network by adjusting synaptic weights and inputs. They show that the spectral slope is only predictive of the E-I ratio under specific conditions, as in most cases the slope does not provide reliable information about the underlying excitation-inhibition ratio. As the authors correctly highlight, the E-I ratio is an important parameter for characterizing a brain network, and the ability to estimate it non-invasively would be a valuable advancement. Therefore, the paper's focus on exploring the limitations of spectral slope as a proxy for E-I ratio is highly relevant and timely. However, I have several concerns regarding the methods used, which I believe should be addressed for a more unambiguous interpretation of the results.

Overall, while open to the possibility that the main claim of the authors is correct (i.e. the relationship between E-I ratio and the slope of the PSD only holds under certain conditions), it is my opinion that, in its current state, this claim is mainly a consequence of a series of questionable parameter choices by the authors, and not something that is intrinsic to the (lack of a) relationship between E-I ratio and PSD slope itself.

Main comments:

1. The paper's central question is the relationship between the slope of aperiodic activity and the E-I ratio. However, the exact definition of E-I ratio is not provided. According to lines 40-45, the E-I ratio in the network is modulated by changing the synaptic weights and inputs ("To address this question, we simulated network activity under different levels of EI-balance (as determined by synaptic weights and inputs) and measured the slope of population activity spectrum."). However, this description does not explain how to calculate the E-I ratio for a single network simulation. For example, in Fig.4, it is unclear what the E-I ratio is for the network realizations without and with inhibitory input to the GPe (represented by the black dots and red arrowheads, respectively). One possible solution would be to define the E-I ratio as the ratio of excitatory to inhibitory conductance (following Gao et al. 2017, where "E:I ratio is defined as the ratio of mean excitatory conductance to mean inhibitory conductance over the simulation time") and then investigate the relationship between this ratio and the slope of aperiodic activity for both models. This would simplify the interpretation of the results and allow for comparisons with existing literature.

2. I have several questions concerning the neocortical model:

- The current study includes both AMPA and NMDA receptors as excitatory components and inhibitory GABA receptors, which deviates from commonly used models. Could the authors elaborate on the reasoning behind this choice?
- A major concern is that both AMPA and NMDA dynamics are modeled using the same general equation (Equation 4), with the only difference being their respective time constants (AMPA: 3 ms, NMDA: 30 ms). However, the model does not seem to incorporate the voltage-dependent property of NMDA receptors, meaning that NMDA current requires depolarization of the postsynaptic cell to open. Without this nonlinearity, NMDA contributions may be vastly overestimated, thereby affecting network dynamics and the interpretation of results. Could the authors clarify whether NMDA voltage dependence was incorporated in the model? If not, how might this simplification influence the findings? Additionally, it would be helpful to clarify how the NMDA and AMPA time constants were selected (e.g., were they based on experimental data?). This is a crucial comment, as the E-I ratio being captured by the slope of the LFP depends on the fact that excitatory currents (mainly

simulated just with AMPA receptors) have a shorter decay constant than inhibitory currents. If the authors vastly overestimate the contribution of NMDA currents (with their slow temporal decay), it is not surprising that the relationship between E-I ratio and the slope of the LFP fades. Experimental work on miniature or evoked synaptic currents puts the AMPA/NMDA ratio in the 1:5 range (Watt et al., 2000, *Neuron*,) 1:4 range (Chaelon et al., 2003 *J. Neurophys.*), and 1:5 to 1:10 range (Watt et al., 2004 *NatNeuro*). Can the authors compute the ratio of AMPA/NMDA currents in their model and, if it significantly deviates from this range, either change the parameter range in their model or explicitly justify their choice?

- The Methods section does not include details on connection probabilities or the values of g_{max} .

- The study varies the fraction of excitatory synapses that exclusively target NMDA receptors. Could the authors comment on the rationale behind this approach, e.g., whether it is informed by experimental evidence? Additionally, this ratio is varied from 5 to 95%, well beyond the physiological range (around 50/50). What is the rationale for this?

- The design of the STN-GPe network is only briefly outlined, with reference to a previously published paper ([16]) for a more detailed description, which makes it challenging to fully understand its specifics based on the methods section alone. In addition, there are several differences between the model presented in the current study and the one in [16]:

- In [16], the neuron model used is the State-dependent Stochastic Bursting Neuron (SSBN), while in the current study, the model is LIF. Could the authors clarify why this change was made?

- In [16], the synaptic weights and delays are as follows: STN-GPe 1.2nS, 6ms; GPe-STN -0.8nS, 6ms; GPe-GPe -0.7nS, 3ms. In contrast, in the current study, the delays are 5ms, 5ms, and 2ms (1ms shorter than in [16]). The weight range for STN-GPe is now [0.8, 1.2, 1.5, 1.6, 1.8], which includes values from [16], while the range for GPe-STN is [-1.7, -1.5, -1.3, -1.2, -1.1] and does not include the value from [16] (-0.8nS). Could the authors comment on choosing these specific values and ranges? Were these values informed by experimental data, for example?

- Synapse model and connection probabilities are not included in the Methods. Could the authors clarify this point as well?

3. The manner in which LFP is derived from excitatory and inhibitory currents in the Mazzoni paper relies on a number of assumptions that only hold for cortical neurons. The same could also be said more generally for the manner in which LFP is generally interpreted also in experimental contexts. I am not therefore sure how this translates to a brain area that has a completely different architecture such as the basal ganglia (no layers but a nuclear structure). Overall, I think that the study would benefit by focusing more on the cortical model and less on the basal ganglia model, whose significance to the main question of the study is a bit dubious.

4. The authors use the FOOOF method for spectral parameterization. As the technique is central to the paper's findings and is sensitive to the parameter choice, I have the following suggestions to improve the transparency and rigor of the analysis:

- Reporting goodness-of-fit metrics (e.g., R^2 and model error that FOOOF already calculates) for the fits. Additionally, it would be helpful to specify the criteria for including/excluding fits (e.g., only include fits with $R^2 \geq 0.95$).

- Assessing the stability of the results (slope estimates) across different parameter choices. One possible approach would be to use different frequency ranges (in addition to the 50-150 Hz range). As a suggestion for possible frequency ranges, I recommend testing 20-Hz-wide frequency windows within the 30(50)-100 Hz range. This is based on Gao et al. (2017), which showed that the correlation between E-I and PSD slope declines with increasing frequency.

5. In Fig. 5, the authors compare the effect of different network parameters on network state and transition. Including statistical analysis would help better quantify these differences and provide stronger support for the conclusions.

Additionally, it may be helpful to complement the box-and-whiskers plot with other visualizations that display individual data points or distributions, such as violin plots, density plots, KDE plots, scatter plots, or alluvial diagrams.

6. In lines 30-35, the authors wrote that "The spectral estimate of (relative) EI-balance implicitly assumes that the inputs to the network are both uncorrelated and asynchronous", yet the reference is missing. In addition, the authors introduce this assumption (network inputs are uncorrelated and asynchronous) but do not elaborate on why correlated or oscillatory activity might corrupt the validity of spectral slope as a readout of E-I ratio. I suggest expanding the discussion of these limitations by including relevant references or empirical evidence (e.g., specific features in electrophysiological activity that bias slope estimates).

7. In lines 255-260, the authors wrote, "When we consider an unconnected population of neurons driven by excitatory and inhibitory spikes, indeed spectral slope is determined by the ratio of excitatory and inhibitory inputs." I believe it would be helpful to include those results in the paper.

8. The LFP was estimated from only 10 excitatory neurons. Could the authors clarify why this specific number was chosen?

9. The main scope of the paper is to explore the relationship between E-I ratio and the slope of the PSD. Yet, these two variables are never directly plotted against each other. I suggest adding this simple plot instead of some of the plots on figures 4-6-7 that are only tangentially related to the main question of the study.

10. In several instances, the authors presume that the reader has familiarity with the NEST environment, as not enough details are given on how the model works. For instance, the authors write: "To model AMPA and NMDA type synaptic dynamics we used the multi compartment neuronal model (iaf_cond_alpha_mc from NEST simulator v3.8.0). Different ports were assigned to NMDA, AMPA, and GABA receptor, enabling synapses to be categorized based on their input type, i.e., inhibitory GABAergic or excitatory glutamatergic." Several key details are missing. What are the different compartments in the model? In what do the compartments differ? How are the compartments to each other? What are the consequences of the 3 types of synapses being in different compartments (if I understand the text correctly)?

11. I believe that discussing experimental work linking the spectral slope with the E-I ratio would place the paper's findings

in a broader context. For instance, the following papers have investigated this relationship: Miller et al., 2009; Colombo et al., 2019; Lendner et al., 2020, 2023; Waschke et al., 2021; Trakoshis et al., 2020; Chini et al., 2022; Kozhemiako et al., 2022; Ameen et al., 2024; Diehl and Redish, 2024.

The papers by Trakoshis and Chini are particularly relevant for this manuscript for two reasons: 1) they explicitly experimentally tested the relationship between E-I ratio and inhibition; 2) they replicated the findings from Gao et al., 2017 using a LIF model similar to the one employed in this manuscript.

Minor comments:

Figure 3 no units for Simulation time and Frequency axis

Figure 4D is unclear what is "basal state"

Table 4: two times "Time constant of excitatory synaptic conductance"

lines 20-25: "Non-invasive signals such as electroencephalogram (EEG) and magneto-encephalogram (MEG) are routine acquired from human subjects" - routinely

lines 30-35: "his argument is based on that fact that typically inhibitory synapses are slower than excitatory ones and population activity is largely associated with trans-membrane currents induced by synaptic inputs" - on the fact

lines 215-220: "A lack of clear relationship between the slope of population activity and spectrum excitation-inhibition balance" - "the" and "and" were missing

lines 240-245: the word "comparing" in "Gao et al. [12] have suggested that relative EI-balance information can be obtained by comparing the slope of the spectrum of population activity such as EEG/MEG." is somewhat misleading. I suggest using estimating or inferring.

lines 245-250: the sentence "Essentially what Gao et al. [12] are suggesting is that there is a systematic relationship between the spectral slope and underlying network parameters which determine the EI balance in some way." is quite vague. The phrase "in some way" is imprecise, and it is unclear which network parameters are meant.

Reviewer #2

(Remarks to the Author)

All comments are provided in the attachment. Please check the following attachment.

Reviewer #3

(Remarks to the Author)

Excitatory-inhibitory (E-I) spiking neuronal networks are known to exhibit complicated dynamics. The authors explored an interesting aspect of such models – whether and how the slope of the power spectrum density (PSD) of the local field potential (LFP) depends on the E-I balance properties of the network. This question would be of interest to the researchers studying macroscopic brain signals (such as human EEG). In those researches, the PSD properties are usually important indicators of the underlying brain states, which in turn originated from the underlying E-I neural circuit properties.

Specifically, they explored two models. One model is the STN-GPe network, where they tune the strength of background inputs and the strength of the synaptic connectivity. Another model is the Neocortical network, where they tune the strength of background inputs, fraction of E synapses targeting NMDA receptors, ratio of I to E synaptics weights. The change of these parameters would alter the degree of inhibition of the network (and thus its E-I balance). They focused on analyzing how the oscillation index and the PSD slope of the network alter with the change of parameters, as well as the mutual dependence of these two indexes. Overall, they found that the PSD slope changes in an intricate manner when altering the parameters and could not be simply attributed to the change of E-I balance. Thus, more caution may be needed in interpreting previous literature that relates PSD slope with E-I balance.

I have the following comments:

1. Fig 1 presented a diagram to illustrate how the E-I balance influence the slope of the PSD. Here, the idea is not very clear to me. First, panel B contrasts the LFP difference between the E-dominant and I-dominant cases. What is the major difference here? Second, for what reason this difference lead to different slope as shown in panel C? It would be helpful to append more explanation regarding these.
2. Could the authors explain a bit the reason of setting the delay parameters? In model 1, the delay of I-to-I connection is set

to 2 ms while others are set to 5 ms. In model 2, the delay of I-to-I connection is set to 5 ms while others are set to 2 ms. Is there any reason for such difference?

3. The LFP of the network is defined through averaging the conductance of 10 excitatory neurons. In E-I network, the activities of neurons can exhibit certain heterogeneity (e.g., some neurons spike more while some are more silent), especially when the network connection is constructed with some randomness (which is the case of the present study). I wonder whether there could be some fluctuation for defining LFP in this way (e.g., because of the small number of the chosen neurons for measuring the LFP). Overall, the authors could not find clear determinants governing the slope of the PSD of the LFP. Could it be partially due to the fluctuation in defining LFP?

4. On page 10, the line just under Fig 4 caption: '... we mean excitatory and inhibitory conductances.' This 'mean' should be another word?

5. On page 12, last paragraph: 'When we consider an unconnected population of neurons driven by excitatory and inhibitory spikes, indeed spectral slope is determined by the ratio of excitatory and inhibitory inputs.' Could you give more explanation about why this is the case? Is there any reference/literature about a more general question: what determines the slope of the PSD of an oscillatory signal?

For the reproducibility, the authors have mentioned all the model parameters, model simulation details and data analysis methods to facilitate the reproduction of their results.

Version 1:

Reviewer comments:

Reviewer #2

(Remarks to the Author)

I think that the authors have adequately addressed the comments made by the reviewers in the revised version of the manuscript. Therefore, I have no further comments. Hence I recommend publication in Communications Biology .

Reviewer #3

(Remarks to the Author)

The authors have made revisions to address my comments. I have the following small suggestions for the authors to further consider:

- Regarding the comment 3, I think the authors should mention in the manuscript in some way to illustrate that estimating the LFP by merely averaging 10 neurons would not cause instability issues.
- Regarding the comment 5, the slope of PSD is related to the noise property of the signal (e.g., pink noise), it would be helpful if the author could discuss the types of noise a EI network can generate and how this may relate to the network structural or dynamic properties.

Dear Editor,

We thank you for the opportunity to revise the manuscript. We also thank the reviewers for their constructive feedback.

In the following we provide the point-by-point reply. The reviewers' comments are in the black and our reply is in blue. In the manuscript all the new additions are marked in blue.

sincerely

Arvind on behalf of the co-authors

Reviewer #1 (Remarks to the Author):

In this study, Chakravarty and colleagues investigate whether the spectral slope of aperiodic activity of a power spectrum density (PSD) can indicate the excitation-inhibition ratio in neural networks, with a steeper slope indicating higher inhibition. To explore this, they use two spiking neural network models—one representing the basal ganglia and the other an arbitrary cortical area. They systematically vary the level of inhibition in the network by adjusting synaptic weights and inputs. They show that the spectral slope is only predictive of the E-I ratio under specific conditions, as in most cases the slope does not provide reliable information about the underlying excitation-inhibition ratio. As the authors correctly highlight, the E-I ratio is an important parameter for characterizing a brain network, and the ability to estimate it non-invasively would be a valuable advancement. Therefore, the paper's focus on exploring the limitations of spectral slope as a proxy for E-I ratio is highly relevant and timely. However, I have several concerns regarding the methods used, which I believe should be addressed for a more unambiguous interpretation of the results. Overall, while open to the possibility that the main claim of the authors is correct (i.e. the relationship between E-I ratio and the slope of the PSD only holds under certain conditions), it is my opinion that, in its current state, this claim is mainly a consequence of a series of questionable parameter choices by the authors, and not something that is intrinsic to the (lack of a) relationship between E-I ratio and PSD slope itself.

We thank the reviewer for very helpful suggestions and constructive feedback. In the following we try our best to convince the reviewer that the results are not just a consequence of model parameters.

Main comments:

1. The paper's central question is the relationship between the slope of aperiodic activity and the E-I ratio. However, the exact definition of E-I ratio is not provided. According to lines 40-45, the E-I ratio in the network is modulated by changing the synaptic weights and inputs ("To address this question, we simulated network activity under different levels of EI-balance (as determined by synaptic weights and inputs) and measured the slope of population activity spectrum."). However, this description does not explain how to calculate the E-I ratio for a single network simulation. For example, in Fig.4, it is unclear what the E-I ratio is for the network realizations without and with inhibitory input to the GPe (represented by the black dots and red arrowheads, respectively). One possible solution would be to define the E-I ratio as the ratio of excitatory to inhibitory conductance (following Gao et al. 2017, where "E:I ratio is defined as the ratio of mean excitatory conductance to mean inhibitory conductance over the simulation time") and then investigate the relationship between this ratio and the slope of aperiodic activity for both models. This would simplify the interpretation of the results and allow for comparisons with existing literature.

We agree with the reviewer about not exactly defining EI-ratio. Now we have defined the EI ratio as we have used and it is exactly what the reviewer has suggested. See line 181 in the Methods section.

We implicitly assumed this definition. For instance, already in the Figure 6 we had plotted the change in the oscillation index and slope of the aperiodic activity in the 2D space spanned by excitatory and inhibitory conductances.

However, now following the suggestion for each network we estimated mean excitatory and inhibitory conductance and rendered the slope of the aperiodic activity against the EI-ratio. In the baseline state—where no extra inhibitory input was provided to the STN-GPe network we did not find any consistent or monotonic relationship between the EI-ratio and the exponent values (green dots in Fig. 1 A in this document).

Furthermore, the changes in network dynamics induced by inhibition also failed to reveal any consistent relationship between these two factors (red dots in Fig. 1 A). Same was true for the neocortical model where

we provided extra inhibition by increasing the strength of the inhibitory connections (g) (see Fig. 1 B in this document). However, in a few specific configurations, we did observe that the exponent values decreased with the EI-ratio (lines with negative slope in the Fig. 1 A-B). This suggests that under certain network states or within a specific range of parameters, the network may exhibit a relationship between the EI-ratio and the exponent values.

We have now added Fig. 9 in the revised manuscript to show this observations. (See lines 311-326).

Figure 1: **Analysis of the relationship between the EI-ratio and the exponent values—derived using the FOOOF algorithm applied to the PSD of the total conductance time series.** **A.** For STN-GPe network. Green dots represent baseline states (without inhibition), while red dots correspond to network outputs under 600 Hz inhibitory input. **B.** Same as in A but for the neo-cortical network.

2. I have several questions concerning the neocortical model: - The current study includes both AMPA and NMDA receptors as excitatory components and inhibitory GABA receptors, which deviates from commonly used models. Could the authors elaborate on the reasoning behind this choice?

Most synapses in the brain have both AMPA and NMDA type receptors and in fact there seems to be a gradient of NMDA/AMPA ratio in the neocortex (Wang XJ, Nature Rev Neuroscience 2020). It is somewhat curious that often in network model this biological fact is ignored. And both AMPA and NMDA currents should contribute to the LFP. Therefore, we were interested in how NMDA/AMPA ratio may affect the slope of the population activity.

- A major concern is that both AMPA and NMDA dynamics are modeled using the same general equation (Equation 4), with the only difference being their respective time constants (AMPA: 3 ms, NMDA: 30 ms). However, the model does not seem to incorporate the voltage-dependent property of NMDA receptors, meaning that NMDA current requires depolarization of the postsynaptic cell to open. Without this nonlinearity, NMDA contributions may be vastly overestimated, thereby affecting network dynamics and the interpretation of results. Could the authors clarify whether NMDA voltage dependence was incorporated in the model? If not, how might this simplification influence the findings? Additionally, it would be helpful to clarify how the NMDA and AMPA time constants were selected (e.g., were they based on experimental data?). This is a crucial comment, as the E-I ratio being captured by the slope of the LFP depends on the fact that excitatory

currents (mainly simulated just with AMPA receptors) have a shorter decay constant than inhibitory currents. If the authors vastly overestimate the contribution of NMDA currents (with their slow temporal decay), it is not surprising that the relationship between E-I ratio and the slope of the LFP fades. Experimental work on miniature or evoked synaptic currents puts the AMPA/NMDA ratio in the 1:5 range (Watt et al., 2000, Neuron,) 1:4 range (Chaelon et al., 2003 J. Neurophy), and 1:5 to 1:10 range (Watt et al., 2004 NatNeuro). Can the authors compute the ratio of AMPA/NMDA currents in their model and, if it significantly deviates from this range, either change the parameter range in their model or explicitly justify their choice?

We agree with the reviewer about our implementation of the NMDA receptors. We did not explicitly model Mg block in the NMDA synapse dynamics. Thus, we might be overestimating the NMDA component. However, we are using point neurons, and given the inputs, the membrane is always depolarized to be close to spike threshold. In such a setting, some Mg block will be removed and NMDA currents will be there for each incoming spike. Therefore, given the point neuron model the simplification makes sense. When synapses are on dendrites far away from the soma, then they have a different membrane potential than soma and Mg block has to be explicitly modeled. In our current model adding Mg block will not change anything and extending the model to have neurons with extended dendritic arbors is clearly beyond the scope of the work. Nevertheless, we accept this limitation and have discussed this in the Methods section. See lines 101-106. In parallel to be sure that our results are not simply because of overestimation of NMDA current we simulated the network model without NMDA component in the synapses (Fig. 4) – as is more common in the literature. Even in this setting we did not find a clear relationship between EI-ratio and slope of the aperiodic activity (except in some small region in the parameter space). We have added this result in the Fig. 7 (top row) in the revised manuscript.

- The Methods section does not include details on connection probabilities or the values of g_{max} .

While we did not explicitly state the connection probabilities, we have provided the number of incoming synaptic connections for both the STN-GPe and neocortical networks. The connection probabilities were taken from Kumar et al. 2011 [6]. For the STN-GPe network, the number and synaptic strength (g_{max}) of incoming excitatory or inhibitory connections for STN and GPe neurons are listed in the Table 1 in the revised manuscript. In the neocortical network, which includes distinct AMPA, NMDA, and GABA synapses, the g_{max} values for each synapse type are provided separately in the third column of Table 3. The number of incoming connections (connection probabilities) for both pyramidal neurons and interneurons, listed in the second column of Table 3, were taken from the work of Kumar et al. 2008 [7]. Captions for Tables 1 and 3 have been updated accordingly. (See lines 62-82).

- The study varies the fraction of excitatory synapses that exclusively target NMDA receptors. Could the authors comment on the rationale behind this approach, e.g., whether it is informed by experimental evidence? Additionally, this ratio is varied from 5 to 95%, well beyond the physiological range (around 50/50). What is the rationale for this?

This choice was imposed by the constraints of the NEST simulator. Within the existing setup a synapse could have only one time constant and adding new model in the NEST source code would have been too tedious. In the brain synapses typically have both NMDA and AMPA components with varying degree. LTP often increases the AMPA component, so to an extent our model captures a setting where synapses are either AMPA dominated or NMDA dominated. This is indeed a simplification we had to make given the simulator.

But as noted earlier our key results do not depend on this choice – the key result can be observed even without NMDA component in the synapse (see Figure 7 top row in the manuscript).

Now we have also revised this figure (see Fig. 4 in the response file) to keep the NMDA/AMPA ratio in a smaller range. See Fig. 7 in the revised manuscript.

- The design of the STN-GPe network is only briefly outlined, with reference to a previously published paper ([16]) for a more detailed description, which makes it challenging to fully understand its specifics based on the methods section alone. In addition, there are several differences between the model presented in the current study and the one in [16]:

The details of the two networks namely, the STN-GPe and the neocortical networks are now expanded in the Methods section. The parameters for both networks were specifically tuned to test our hypothesis regarding the excitation-inhibition EI (E/I)-ratio.

- In [16], the neuron model used is the State-dependent Stochastic Bursting Neuron (SSBN), while in the current study, the model is LIF. Could the authors clarify why this change was made?

The SSB model was used in Bahuguna et al. [1] to study the effects of spike bursting. In this study we are not interested in spike bursting or pathological activity associated with Parkinson's disease. Our goal was to vary the EI-ratio and there are only two options in the model: change the synaptic connections or the inputs.

- In [16], the synaptic weights and delays are as follows: STN-GPe 1.2nS, 6ms; GPe-STN -0.8nS, 6ms; GPe-GPe -0.7nS, 3ms. In contrast, in the current study, the delays are 5ms, 5ms, and 2ms (1ms shorter than in [16]). The weight range for STN-GPe is now [0.8, 1.2, 1.5, 1.6, 1.8], which includes values from [16], while the range for GPe-STN is [-1.7, -1.5, -1.3, -1.2, -1.1] and does not include the value from [16] (-0.8nS). Could the authors comment on choosing these specific values and ranges? Were these values informed by experimental data, for example?

We would like to point out that in most cases such data about synaptic strengths are not available as often in experimental settings synaptic strength are measured *in vitro*. However, *in vivo* synaptic strengths could be quite different from those *in vitro* due to the high conductance state of neurons. So the strategy used in computational models is to choose ball-park values to get the network operating in an activity range which is consistent with the data – in this case firing rates of the STN/GPE neurons and population oscillations. Once we get to a physiological range of activity we vary the synaptic strengths to test the parameter sensitivity of the results. This is a standard practice in computational neuroscience.

We have chosen this range because this allows us to see both healthy and unhealthy state of the network and the parameter sampling is sufficiently dense to see the transition between the two states. We have clarified this point in the Methods section (See lines 141-149).

We identified a parameter set that produces a state space with distinguishable oscillatory and non-oscillatory regimes. To explore differences in the aperiodic components, we subsequently employed a slightly modified parameter set (different from -0.8nS) that generates a state space exhibiting such variations.

- Synapse model and connection probabilities are not included in the Methods. Could the authors clarify this point as well?

We have now provided the number of incoming synaptic connections for both the STN-GPe and neocortical networks (See Tables 1 and 3). The connection probabilities were taken from Kumar et al. 2011 (the reference has been in line 68 in the revised manuscript). For the STN-GPe network, the number and synaptic strength (g_{\max}) of incoming excitatory or inhibitory connections for STN and GPe neurons are listed in the Table 1 in the revised manuscript. In the neocortical network, which includes distinct AMPA, NMDA, and GABA synapses, the g_{\max} values for each synapse type are provided separately in the third column of Table 3. The number of incoming connections (connection probabilities) for both pyramidal neurons and interneurons, listed in the second column of Table 3, were taken from the work of Kumar et al 2008 (see line 79). Captions for Tables 1 and 3 have also been updated accordingly.

In our models, all synapses are implemented as conductance based synapses (line 68 and newly added line 81). To better explain the dynamics of conductance-based synapses, we have added a dedicated section titled *Synapse Model* in the *Methods* (See Lines 84-106), where the underlying formulation and implementation details are described.

3. The manner in which LFP is derived from excitatory and inhibitory currents in the Mazzoni paper relies on a number of assumptions that only hold for cortical neurons. The same could also be said more generally for the manner in which LFP is generally interpreted also in experimental contexts. I am not therefore sure how this translates to a brain area that has a completely different architecture such as the basal ganglia (no layers but a nuclear structure). Overall, I think that the study would benefit by focusing more on the cortical model and less on the basal ganglia model, whose significance to the main question of the study is a bit dubious.

We respectfully disagree with the reviewer about the significance of the BG model for this study. The question we are addressing (the relationship between the spectrum slope and EI-ratio) may depend on the choice of network architecture and not just on the strength of excitatory and inhibitory synapses. Therefore, we wanted to study at least two different architectures. In the brain we encounter three main motifs: EE-EI-IE-II motif: where there is mutual and within connectivity among both excitatory and inhibitory neurons e.g. in the neocortex, EI-IE-II motif: where E-E recurrent connections are missing e.g. in CA1 and STN-GPe and II motif: where there are only inhibitory neurons e.g. in the striatum. In this study we have considered the two main motifs EE-EI-IE-II motif and EI-IE-II motif. With the STN-GPe model we have verified that results from EI-IE-II motif are consistent with those from EE-EI-IE-II motif. Moreover, the STN-GPe motif allows us to check the results when there is a high state transition from non-oscillatory to oscillatory states. Finally, results from STN-GPe have relevance to study Parkinson's disease from the perspective of EI-ratios. We have clarified this in the opening of the results section. (See lines 186-193).

Indeed basal ganglia does not have layers like in the neocortex but that does not mean that there is no LFP in these regions. One simple way to get effective dipole – which will then produce field – is to consider that not whole dendritic arbor of the STN or GPe neurons is activated equally at any given time (personal communication Gaute Einevoll).

4. The authors use the FOOOF method for spectral parameterization. As the technique is central to the paper's findings and is sensitive to the parameter choice, I have the following suggestions to improve the transparency and rigor of the analysis:

- Reporting goodness-of-fit metrics (e.g., R^2 and model error that FOOOF already calculates) for the fits. Additionally, it would be helpful to specify the criteria for including/excluding fits (e.g., only include fits with $R^2 \geq 0.95$).

We thank the reviewer for the suggestion. We have carefully evaluated the results obtained from the FOOOF algorithm, which was applied to the PSD of average conductance. FOOOF was able to fit the data well, with an R^2 of 0.94 ± 0.026 for the STN-GPe network and 0.971 ± 0.0035 for the neocortical network, showing that it can reliably estimate the aperiodic (γ) components. Maximum number of allowed peaks in the PSD within the 50-150 Hz range (post- β range): 16; minimum peak height set to 0.01; peak threshold set to 2; peak width limits set to [0.5, 12.0] and the aperiodic mode was fixed.

We have now added FOOOF parameter set and its goodness of fit in the Data Analysis section, lines 173-180.

- Accessing the stability of the results (slope estimates) across different parameter choices. One possible approach would be to use different frequency ranges (in addition to the 50-150 Hz range). As a suggestion for possible frequency ranges, I recommend testing 20-Hz-wide frequency windows within the 30(50)–100 Hz range. This is based on Gao et al. (2017), which showed that the correlation between E-I and PSD slope declines with increasing frequency.

To ensure the stability of the results, we ran 25 trials for each parameter set and then estimated the average slope of the aperiodic activity. (see lines 200-201)

5. In Fig. 5, the authors compare the effect of different network parameters on network state and transition. Including statistical analysis would help better quantify these differences and provide stronger support for the conclusions. Additionally, it may be helpful to complement the box-and-whiskers plot with other visualizations that display individual data points or distributions, such as violin plots, density plots, KDE plots, scatter plots, or alluvial diagrams.

We thank the reviewer for this suggestion. We have now updated the Figure 5 with violin plots (see Fig. 2 below and new Figure 5 in the revised manuscript). Now to assess statistical significance, we performed a t-test for each of the parameters shown in the violin plots Fig. 2 for the STN-GPe network.

6. In lines 30-35, the authors wrote that “The spectral estimate of (relative) E-I-balance implicitly assumes that the inputs to the network are both uncorrelated and asynchronous”, yet the reference is missing. In addition, the authors introduce this assumption (network inputs are uncorrelated and asynchronous) but do not elaborate on why correlated or oscillatory activity might corrupt the validity of spectral slope as a readout of E-I ratio. I suggest expanding the discussion of these limitations by including relevant references or empirical evidence (e.g., specific features in electrophysiological activity that bias slope estimates).

We thank the reviewer for raising this concern. Indeed we should have elaborated on this point. We have now expanded the discussion to explain this better – it does not really fit in the introduction. Briefly, when spikes are correlated, each incoming spike arrives with other spikes. Therefore from the perspective of the postsynaptic neuron this situation is akin to have synaptic strength increased by a factor dependent on the degree of correlation. So increasing correlation in excitatory population for instance may reduce the slope of the population activity spectrum (see Fig. 5 in the response document). We have added this new figure in the supplementary methods. It is well known that changes in the excitation and inhibition in a network not only changes the firing rates but also the correlations. Therefore, in a network it is non-trivial to find relationship between E-I-ratio and spectral slope. See lines 299-308 in Discussion.

Figure 2: **Violin plots to analyze effect of network parameters in shaping the network state and its transition when GPe is inhibited.** For a meaningful comparison we normalized each parameter using the formula $\frac{x - \min(x)}{\max(x) - \min(x)}$, where x comprises ($wt_{STN-GPe}$, $wt_{GPe-STN}$, bg_{STN}^{ex} , bg_{GPe}^{ex}). **(A)** Distribution of four parameters when the network was either in low-oscillation or high-oscillation regimes. This refers to the states marked with black dots in the Fig. 4 A-C. **(B)** The distribution of parameters during various types of inhibition-induced transitions, namely low to low, low to high, and high to high oscillation zones. **(C)** Same as in **B** Distribution of parameters when GPe inhibition resulted in transition to a smaller and higher exponent value from their baseline. This panel was made using data shown in Fig. 4 C in the revised manuscript. “***” denotes significance level (p-value): green “***” for $p < 0.001$, green “**” for $p < 0.01$, green “*” for $p < 0.05$. A red “*” indicates a non-significant result ($p > 0.05$).

7. In lines 255-260, the authors wrote, “When we consider an unconnected population of neurons driven by excitatory and inhibitory spikes, indeed spectral slope is determined by the ratio of excitatory and inhibitory inputs.” I believe it would be helpful to include those results in the paper.

We think that this is not necessary as it is already there in the Gao paper. We now added that reference. However, as explained in the previous point, we have also expanded the discussion to address this comment (lines 300-309 in Discussion).

8. The LFP was estimated from only 10 excitatory neurons. Could the authors clarify why this specific number was chosen?

Since the network is largely homogeneous—apart from the threshold voltages of STN and GPe neurons, which were randomly varied within ± 2 mV. In such a setting averaging over the whole neuron population would yield results identical to what we have reported with 10 neuron. To verify this we estimated the LFP from 10 and 100 neurons and computed the oscillation index and exponent of the spectral slope (Fig. 3) (see lines 156-160 in Methods section). Unless the reviewer strongly recommends it, we do not see a good reason to include these observations in the manuscript.

Figure 3: **Comparison of OI_{avg} and λ_{avg} values derived by averaging of total conductances /LFPs across 10 neurons and 1000 neurons in the STN population.** (A) Each dot represents OI_{avg} measured for 10 neurons (x-axis) or 1000 neurons (y-axis). This was done for a subset of 250 network configuration parameters selected randomly from a total of 625 network configurations. As is evident from this data, OI_{avg} estimates from 10 neurons is highly correlated with that estimates from 1000 neurons. (B) Same as in panel A but for λ_{avg} . These data justify our choice of using 10 neurons for LFP generation in the manuscript. Note that same result holds for the neocortical networks.

9. The main scope of the paper is to explore the relationship between E-I ratio and the slope of the PSD. Yet, these two variables are never directly plotted against each other. I suggest adding this simple plot instead of some of the plots on Figures 4-6-7 that are only tangentially related to the main question of the study.

This concern has been addressed in the reply to the first major concern raised by the reviewer (See Fig. 9 A-B in the revised manuscript and our reply earlier on 2).

10. In several instances, the authors presume that the reader has familiarity with the NEST environment, as not enough details are given on how the model works. For instance, the authors write: "To model AMPA and NMDA type synaptic dynamics we used the multi compartment neuronal model (*iaf_cond_alpha_mc* from NEST simulator v3.8.0). Different ports were assigned to NMDA, AMPA, and GABA receptor, enabling synapses to be categorized based on their input type, i.e., inhibitory GABAergic or excitatory glutamatergic." Several key details are missing. What are the different compartments in the model? In what do the compartments differ? How are the compartments to each other? What are the consequences of the 3 types of synapses being in different compartments (if I understand the text correctly)?

In NEST, we employed a multi-compartmental neuron model *iaf_cond_alpha_mc* to accommodate AMPA, NMDA, and GABA synapses. It consists of three compartments: soma, proximal, and distal. In our implementation, however, we employed a point neuron modeling approach. This decision was based on the flexibility offered by NEST's multi-compartmental framework, which allows for compartment-specific tuning of parameters such as synaptic time constants and conductances for both excitatory and inhibitory inputs. Given that NMDA, AMPA, and GABA receptors each have distinct time constants and conductance properties, we leveraged this feature to model the functional role of compartments through receptor-specific dynamics.

In our implementation, GABA receptors were associated with the soma compartment, while AMPA and

NMDA receptors were assigned to proximal and distal dendritic compartments, respectively. The synaptic time constants, leak conductance, and capacitance values were tuned to ensure that each pyramidal neuron maintained a firing rate about 0.68-1.2 Hz, while also ensuring that AMPA synapses exhibited faster dynamics compared to NMDA synapses. To better explain this, we have added a dedicated section titled *Synapse Model* in the Methods (line 84).

11. I believe that discussing experimental work linking the spectral slope with the E-I ratio would place the paper's findings in a broader context. For instance, the following papers have investigated this relationship: Miller et al., 2009; Colombo et al., 2019; Lendner et al., 2020, 2023; Waschke et al., 2021; Trakoshis et al., 2020; Chini et al., 2022; Kozhemiako et al., 2022; Ameen et al., 2024; Diehl and Redish, 2024. The papers by Trakoshis and Chini are particularly relevant for this manuscript for two reasons: 1) they explicitly experimentally tested the relationship between E-I ratio and inhibition; 2) they replicated the findings from Gao et al., 2017 using a LIF model similar to the one employed in this manuscript.

We thank the reviewer for pointing us to these papers. Some of these we have now included in the manuscript. We note that in general our results do not match with those described in Trakoshis et al. [8] and Chini et al. [3]. Both studies showed that an increase in inhibitory synaptic strength results in a corresponding increase in spectral slope. But we found that this may only hold for a restricted range of parameters. There are many more examples in our simulations where spectral slope decreased when inhibition was increased (See the new Figure 9). See line 323 in discussion.

Minor comments:

Figure 3 no units for Simulation time and Frequency axis.

Thanks for raising this issue. We have corrected Figure 3 in the revised manuscript.

Figure 4D is unclear what is "basal state"

Thanks for raising this issue. We have removed the phrase and also updated the caption of Figure 4D.

Table 4: two times "Time constant of excitatory synaptic conductance"

Thanks. We have corrected the typo.

lines 20-25: "Non-invasive signals such as electroencephalogram (EEG) and magneto-encephalogram (MEG) are routinely acquired from human subjects" - routinely

Thanks. We have corrected the word.

lines 30-35: "his argument is based on that fact that typically inhibitory synapses are slower than excitatory ones and population activity is largely associated with trans-membrane currents induced by synaptic inputs" - on the fact

Thanks. We have corrected the typo.

lines 215-220: "A lack of clear relationship between the slope of population activity and spectrum excitation-inhibition balance" - "the" and "and" were missing

Thank you for noticing this. We have corrected this typo.

Figure 4: **Analysis of the relationship between the network dynamics (OI_{avg} , λ_{avg}), in a neocortical network with AMPA-only synapses (no NMDA) with respect to external Poisson input (η) and also with EI-ratio (g). Here, AMPA is connected to proximal and NMDA to distal. A. Heatmap for OI_{avg} . B. Heatmap for λ_{avg} . Each row represents the network with specific ratio of NMDA and AMPA current in excitatory synapses.**

lines 240-245: the word "comparing" in "Gao et al. [12] have suggested that relative EI-balance information can be obtained by comparing the slope of the spectrum of population activity such as EEG/MEG." is somewhat misleading. I suggest using estimating or inferring.

We agree with the reviewer and have changed the wording.

lines 245-250: the sentence "Essentially what Gao et al. [12] are suggesting is that there is a systematic relationship between the spectral slope and underlying network parameters which determine the EI balance in some way." is quite vague. The phrase "in some way" is imprecise, and it is unclear which network parameters are meant.

We agree and we have now revised this line. See lines 286-288 in Discussion section.

Reviewer #2 (Remarks to the Author):

All comments are provided in the attachment.

The following is taken from the attachment.

This article explores that whether the slope of the spectrum can predict the ratio of excitatory and inhibitory synaptic conductance. The authors claimed that only in a few network configurations an increase in the inhibitory synaptic strength was associated with a corresponding increase in the spectral slope. In most cases, an increase in the spectral slope could be due to both an increase or a decrease in strength of inhibitory inputs. The main results of this investigation suggested that the change in the spectral slope is not a reliable predictor of excitatory-inhibitory balance in terms of network parameters or synaptic conductance. I recommend reconsidering based on responses to issues raised by reviewers. The model design and description in the paper are sufficient, and implementing alpha type of synaptic models in the EI-neural network of leaky integrate-and-fire neuron is a challenging and meaningful endeavor. The subject aligns well with the journal's scope. However, there are some shortcomings in the manuscript that need to be addressed. Besides a long list of minor changes (given below), I have a few major points of criticism that I ask the authors to address before I can recommend publication in *Communications Biology*.

We thank the reviewer for thoughtful suggestions and constructive feedback.

Major points:

1. Please provide some recent literature on synchronization of excitatory-inhibitory balance network. e.g., Zhu et al., *Nonlinear Dynamics* 112 (9), 7555-7570; Zhou et al., *Chaos, Solitons and Fractals*, 165, p.112891.

In this work, we are not concerned with synchrony in the network. Therefore, we think that these references are not relevant to our work.

2. The network structures in the STN-GPe and neocortical networks are not clearly defined. How can neurons be connected by synapses? Are they all-to-all within or between two populations?

We have revised the Methods section to give further details of the network structure. Briefly, the neurons were connected with fixed probability. All connections were made using conductance-based synapses. See lines 65-100.

3. In the phrase "The term I_{syn} represents the sum of all synaptic inputs." found at the top of Figure 2. does I_{syn} refer to the synaptic currents defined by Eqs 1-3, which are activated by AMPA, NMDA, and GABA receptors? Additionally, do all neurons receive the same number of connections that contribute to I_{syn} ?

The reviewer is right about I_{syn} . But eq. 1-3 only describe individual synaptic currents, therefore we did not reference the equations while describing I_{syn} . In both the networks, all neurons received the same number of inputs, but out-degree of the neurons should follow a binomial distribution. The connections were implemented in NEST using the option 'Fixed In-degree'. See lines 69-70 and 84-100.

4. The phrase "conductance-based synapses" in the sentence "Neurons were connected randomly with a fixed connection probability using conductance-based synapses (see Table 1; for more details, please refer to [16])" is crucial for helping the reader understand the concept. The authors should provide a definition of this term in this section.

We thank the reviewer for raising this issue. This reveals an inconsistency in our description of Methods. Now we have clarified it. To better explain the dynamics of conductance-based synapses, we have added a dedicated section titled *Synapse Model* in the Methods (line 84-106), where the underlying formulation and implementation details are described.

5. How can the authors provide default values for the parameters in Table 2 based on experiments?

We do not have any default values. We have systematically varied each parameter. The parameter ranges were chosen to ensure that the network activity remains within the physiological range experimentally observed. Also synaptic strengths were chosen such that each synapse remains weak. In general all parameter ranges were taken from previous papers which are already referenced in the manuscript. Also we have added a note to give a rationale behind the choice of model parameters. See lines 141-149.

6. what does " $\zeta \in [50 : 50 : 750]$ " mean? It is so blurred

The notation $\zeta \in [50 : 50 : 750]$ denotes that the value of ζ ranges from 50 to 750, with a step size of 50. However, in response to Reviewer 1's comments, we have updated range of ζ . For better understanding, we have updated the expression with $\zeta \in (50 \times i)$ where $i \in [0, 10]$ (see lines 129-130 in Methods). We also included the updated results in the revised manuscript (see Fig. 7).

7. Do 3D and 4D represent three-dimensional and four-dimensional parameter spaces, respectively?

Yes, the reviewer is right. The 4D parameter set refers to four input parameters of the STN-GPe network: the synaptic weight from STN to GPe ($(STN-GPe)_{wt}$), the synaptic weight from GPe to STN ($(GPe-STN)_{wt}$), and the background Poisson-type inputs to STN (STN_{bg}) and GPe (GPe_{bg}) neurons. For the neocortical network, a 3D parameter set was used, involving three network configuration parameters: (1) the ratio ('g') of inhibitory to excitatory synaptic weights, (2) the rate of external excitatory background (Poisson) input (η) to both pyramidal and interneuron populations, and (3) the fraction of excitatory synapses targeting NMDA receptors exclusively (ζ). See lines 116-121, 127-134 in Methods section.

8. The equation $\sum_k p_k = 1$ indicates a summation over all values of k. However, the author claims that k represents frequency. Which interpretation is correct?

We thank the reviewer for noticing this inconsistency. We have now modified the equation 4 in the revised manuscript. (see line 170-172)

9. It is better to move The formula $(x - \min(x))/(\max(x) - \min(x))$ to the main text from The caption of

Figure 5.

We agree that the expression is a bit unusual in the figure caption. But think it is appropriate to keep it in the figure caption. We thought that it would be unnecessary to ask the reader to go to methods and then return to the figure (as this figure is a bit far from the methods).

Minor points: Throughout the manuscript: Some sentences are too long to effectively convey the main concepts.eg "In line with our investigation of the STN-GPe network, our experimental design for the neo-cortical network involved a systematic adjustment of . . . within the pyramidal populations." below the line 105.

We understand and agree with this concern, Therefore, now we have tried to improve on the sentence length.

Reviewer #3 (Remarks to the Author):

Excitatory-inhibitory (E-I) spiking neuronal networks are known to exhibit complicated dynamics. The authors explored an interesting aspect of such models – whether and how the slope of the power spectrum density (PSD) of the local field potential (LFP) depends on the E-I balance properties of the network. This question would be of interest to the researchers studying macroscopic brain signals (such as human EEG). In those researches, the PSD properties are usually important indicators of the underlying brain states, which in turn originated from the underlying E-I neural circuit properties.

Specifically, they explored two models. One model is the STN-GPe network, where they tune the strength of background inputs and the strength of the synaptic connectivity. Another model is the Neocortical network, where they tune the strength of background inputs, fraction of E synapses targeting NMDA receptors, ratio of I to E synapses weights. The change of these parameters would alter the degree of inhibition of the network (and thus its E-I balance). They focused on analyzing how the oscillation index and the PSD slope of the network alter with the change of parameters, as well as the mutual dependence of these two indexes. Overall, they found that the PSD slope changes in an intricate manner when altering the parameters and could not be simply attributed to the change of E-I balance. Thus, more caution may be needed in interpreting previous literature that relates PSD slope with E-I balance.

We thank the reviewer for helpful suggestions and feedback.

I have the following comments:

1. Fig 1 presented a diagram to illustrate how the E-I balance influence the slope of the PSD. Here, the idea is not very clear to me. First, panel B contrasts the LFP difference between the E-dominant and I-dominant cases. What is the major difference here? Second, for what reason this difference lead to different slope as shown in panel C? It would be helpful to append more explanation regarding these.

Figure 1 in the manuscript is a schematic diagram illustrating how excitation and inhibition are expected to influence the power spectrum. Excitatory and inhibitory synapses have different time constants (excitation being faster – shown in Figure 1A). If we assume (and quite correctly in many cases) that LFP is a linear sum of excitatory and inhibitory currents (shown in Figure 1B), slope of the spectrum will depend on whether the LFP is dominated by excitation and inhibition – if there are stronger excitatory synaptic currents then the LFP fluctuations will be faster and therefore more power in the high frequencies and a flatter spectrum (small slope) and vice versa for stronger inhibition. This idea is well explained in the original paper by Gao et al. Therefore, we did not feel the need to elaborate it and simply added a schematic description. However, the main idea is explained in lines 30-43.

2. Could the authors explain a bit the reason of setting the delay parameters? In model 1, the delay of I-to-I connection is set to 2 ms while others are set to 5 ms. In model 2, the delay of I-to-I connection is set to 5 ms while others are set to 2 ms. Is there any reason for such difference?

In this work, all synaptic or transmission delays were taken from previous studies. For the STN-GPe network we referred to Kumar et al. [6], and for the neocortical network we referred to Brunel et al. [2]. In these previous studies delays were chosen partly based on experimental evidence of delays and partly to match the network activity dynamics to the experimentally recorded activity. We did not investigate the effect of delay parameters in this study. However, we acknowledge that delays can play a crucial role in shaping network oscillations and the exponent values in recurrently connected networks. Our previous work suggest that delays

can induce oscillations (see Kim et al. 2021 PRE [5]). How such oscillations affect the spectral slope is not clear and will be investigated in a future study. We have added a line to state this limitation of our work in the discussion section. See lines 335-338

3. The LFP of the network is defined through averaging the conductance of 10 excitatory neurons. In E-I network, the activities of neurons can exhibit certain heterogeneity (e.g., some neurons spike more while some are more silent), especially when the network connection is constructed with some randomness (which is the case of the present study). I wonder whether there could be some fluctuation for defining LFP in this way (e.g., because of the small number of the chosen neurons for measuring the LFP). Overall, the authors could not find clear determinants governing the slope of the PSD of the LFP. Could it be partially due to the fluctuation in defining LFP?

We thank the reviewer for raising this concern. Reviewer 1 also raised a similar concern. Since the network is largely homogeneous – apart from the threshold voltages of STN and GPe neurons, which are randomly varied within ± 2 mV. In such a setting averaging over the whole neuron population would yield results identical to what we have reported with 10 neurons. To verify this, we estimated LFP from 10 and 100 neurons and computed the oscillation index and exponent of the spectral slope (Fig. 3). Unless the reviewer recommends it, we do not see a good reason to include these observations in the manuscript.

4. On page 10, the line just under Fig 4 caption: ‘... we mean excitatory and inhibitory conductances.’ This ‘mean’ should be another word?

We agree that this was confusing. We have now modified the text in the original manuscript (see line 246-248).

5. On page 12, last paragraph: ‘When we consider an unconnected population of neurons driven by excitatory and inhibitory spikes, indeed spectral slope is determined by the ratio of excitatory and inhibitory inputs.’ Could you give more explanation about why this is the case? Is there any reference/literature about a more general question: what determines the slope of the PSD of an oscillatory signal?

The best reference for this is indeed Gao et al. *NeuroImage* 2017 [4]. As explained above, excitatory and inhibitory inputs have different time constants and LFP is a sum of synaptic currents. When we consider unconnected neurons which are receiving Poisson type inputs the situation is straightforward: spectrum of the LFP is a sum of spectra of excitatory and inhibitory currents. Therefore, the spectral slope is determined by the balance between excitatory and inhibitory inputs. This was demonstrated in the work by Gao et al., published in *NeuroImage* in 2017. Their findings demonstrated that spectral features, such as the $1/f$ slope, become flatter as the excitation-to-inhibition (EI) ratio increases (see Figure 1 in Gao. et al *NeuroImage* 2017).

For the reproducibility, the authors have mentioned all the model parameters, model simulation details and data analysis methods to facilitate the reproduction of their results.

No reply needed.

Figure 5: **Effect of input correlations on the aperiodic exponent and the variance of total synaptic conductance (as reflected in the LFP).** **A** Schematic of the correlated inputs to the two neurons. Each neuron received Poisson type excitatory and inhibitory spike trains. Inhibitory inputs to each neuron came from independent sources. To control input correlations we varied the fraction of shared input. Correlation among excitatory inputs was controlled by varying the number of shared Poisson-type spike trains. Input correlation increases monotonically with the degree of shared (presynaptic) inputs between the neurons. **B** Aperiodic component as a function of input sharing. **C** Variance of simulated LFPs as a function of input sharing.

References

- [1] Jyotika Bahuguna, Ajith Sahasranamam, and Arvind Kumar. Uncoupling the roles of firing rates and spike bursts in shaping the stn-gpe beta band oscillations. *PLoS computational biology*, 16(3):e1007748, 2020.
- [2] Nicolas Brunel. Dynamics of sparsely connected networks of excitatory and inhibitory spiking neurons. *Journal of computational neuroscience*, 8:183–208, 2000.
- [3] Mattia Chini, Thomas Pfeffer, and Ileana Hanganu-Opatz. An increase of inhibition drives the developmental decorrelation of neural activity. *elife*, 11:e78811, 2022.
- [4] Richard Gao, Erik J Peterson, and Bradley Voytek. Inferring synaptic excitation/inhibition balance from field potentials. *Neuroimage*, 158:70–78, 2017.
- [5] Christopher M Kim, Ulrich Egert, and Arvind Kumar. Dynamics of multiple interacting excitatory and inhibitory populations with delays. *Physical Review E*, 102(2):022308, 2020.
- [6] Arvind Kumar, Stefano Cardanobile, Stefan Rotter, and Ad Aertsen. The role of inhibition in generating and controlling parkinson's disease oscillations in the basal ganglia. *Frontiers in systems neuroscience*, 5:86, 2011.
- [7] Arvind Kumar, Sven Schrader, Ad Aertsen, and Stefan Rotter. The high-conductance state of cortical networks. *Neural computation*, 20(1):1–43, 2008.
- [8] Stavros Trakoshis, Pablo Martínez-Cañada, Federico Rocchi, Carola Canella, Wonsang You, Bhismadev Chakrabarti, Amber NV Ruigrok, Edward T Bullmore, John Suckling, Marija Markicevic, et al. Intrinsic excitation-inhibition imbalance affects medial prefrontal cortex differently in autistic men versus women. *elife*, 9:e55684, 2020.

Dear Dr. Pan

Thank you very much. We are really thrilled with the decision. Below you will find replies to the two comments from the reviewer #3 and how we have revised the manuscript.

Best wishes

Arvind Kumar on behalf of the authors.

Reviewer #3 (Remarks to the Author):

The authors have made revisions to address my comments. I have the following small suggestions for the authors to further consider:

- Regarding the comment 3, I think the authors should mention in the manuscript in some way to illustrate that estimating the LFP by merely averaging 10 neurons would not cause instability issues.

We have now added the following figure as a supplement to clarify this point (see Lines 356 in the main

Figure 3: **Comparison of OI_{avg} and λ_{avg} values derived by averaging of total conductances /LFPs across 10 neurons and 1000 neurons in the STN population.** (A) Each dot represents OI_{avg} measured for 10 neurons (x-axis) or 1000 neurons (y-axis). This was done for a subset of 250 network configuration parameters selected randomly from a total of 625 network configurations. As is evident from this data, OI_{avg} estimates from 10 neurons is highly correlated with that estimates from 1000 neurons. (B) Same as in panel A but for λ_{avg} . These data justify our choice of using 10 neurons for LFP generation in the manuscript. Note that same result holds for the neocortical networks. (text).

- Regarding the comment 5, the slope of PSD is related to the noise property of the signal (e.g., pink noise), it would be helpful if the author could discuss the types of noise a EI network can generate and how this may relate to the network structural or dynamic properties.

The exact spectrum depends on the activity regime, network structure, neuron and synapse properties. When the network is tuned to operate in an asynchronous-irregular state the spectral power of the noise is by definition $1/f^a$ where $a > 0$. When the network is operating in an oscillating regime, additional peaks are added in the $1/f^a$ spectrum. In some cases when neurons have voltage-gated channels it is possible that neurons do not behave as low-pass filters. In such a setting, population activity generated by the network may have significant deviations $1/f^a$ spectrum. To the best of our knowledge, no one has shown the existence of noise in which spectral power increases with frequency i.e. $a < 0$ (e.g. blue noise) in neural networks. We have mentioned this in the manuscript (see lines 180). We have to admit that a full treatment of this issue is a bit tangential to our manuscript.